

# Impact of biomass burning on pollutants surface concentrations in megacities of the Gulf of Guinea

Laurent MENUT[1], Cyrille FLAMANT[2], Solène TURQUETY[1], Adrien DEROUBAIX[1], Patrick CHAZETTE[3], and Rémi MEYNADIER[2]

[1]Laboratoire de Météorologie Dynamique, Ecole Polytechnique, IPSL Research University, Ecole Normale Supérieure, Université Paris-Saclay, Sorbonne Universités, UPMC Univ Paris 06, CNRS, Route de Saclay, 91128 Palaiseau, France
[2]LATMOS/IPSL, Sorbonne Universités, UPMC Univ Paris 06, UVSQ, CNRS, 75252 Paris, France
[3]LSCE

*Correspondence to:* Laurent Menut, menut@lmd.polytechnique.fr

**Abstract.** In the framework of the "Dynamics-Aerosol-Chemistry-Cloud Interactions in West Africa" (DACCIWA) project, the tropospheric chemical composition in the megacities along the Guinean Gulf is studied using the Weather and Research Forecast and CHIMERE regional models. Simulations are performed for the May-July 2014 period, without and with biomass burning emissions. Model results are compared to satellite data and surface measurements. Using numerical tracer release experiments, it is shown that the fire emissions in Central Africa are impacting the surface aerosol and gaseous species concentrations in the Guinean Gulf cities, such as Lagos (Nigeria) and Abidjan (Ivory Coast). Depending on the altitude of injection of these emissions, the pollutants follow different pathways: directly along the coast or over land towards the Sahel before to be vertically mixed in the convective boundary layer and transported to the south-west and over the cities. In July 2014, the maximum increase in surface concentrations is $\approx 150 \ \mu g \ m^{-3}$ for CO, $\approx 10$ to $20 \ \mu g \ m^{-3}$ for $O_3$ and $\approx 5 \ \mu g \ m^{-3}$ for $PM_{10}$. The analysis of the $PM_{10}$ chemical composition shows that this increase is mainly related to an increase of Particulate Primary Matter and Particulate Organic Matter.

## 1 Introduction

The concentrations of gases and particles are rapidly growing in southern West Africa (SWA), driven by the constant increase of anthropogenic atmospheric emissions. These emissions are linked with car traffic, industries and related gas and oil extraction activities, domestic fires and waste burning, (Marais and Wiedinmyer, 2016). They are proportional to the population which is increasing dramatically in urbanized areas, (Adon et al., 2016). The atmospheric pollution problems are mainly present along the coast of the Gulf of Guinea spanning from Abidjan (Ivory Coast) to Port Harcourt (Nigeria) and occur in the lower few hundred of meters above the surface, in the atmospheric boundary layer (ABL). In addition to this anthropogenic regional pollution, the region is impacted by other important sources especially in the summer, with high emissions of mineral dust from the Sahara and the Sahel to the north and vegetation fires from Central and southern Africa (Real et al., 2010). In the coastal region of SWA mineral dust and biomass burning aerosols are generally observed above the ABL, between 800 and 600 hPa, as the result of long-range transport. Mineral dust is transported from the north in the Saharan air layer (Parker et al., 2005),





(Flamant et al., 2009) and can be mixed downward into the ABL over the Sudanian region (Crumeyrolle et al., 2011). Using a Lagrangian model, Mari et al. (2008) have evidenced that the intrusion of southern hemispheric biomass burning aerosol plumes occurred in the mid-troposphere over the Gulf of Guinea, but did not investigate whether these plumes could impact air quality over urbanized areas of SWA.

The variability of the atmospheric composition and its impact on West African climate and on the health of populations and eco-systems is the purpose of Dynamics-Aerosol-Chemistry-Cloud Interactions in West Africa (DACCIWA) project (Knippertz et al., 2015). In this study, we concentrate on the summer of 2014, which was the focus of one of the dry run exercises conducted in preparation of the field campaign that took place in June/July 2016 (Flamant et al., 2017). The period corresponds to the onset of the West African Monsoon (WAM) when the rainy convective systems migrate from the coastal area along the

Gulf of Guinea to the Sahel (Williams et al., 2010). The months of June and July 2014 were more prone to precipitation at the SWA coast than 2015 and 2016, due to a late monsoon onset. The precipitation and the dynamics associated with the related mesoscale convective systems strongly impact the vertical distribution of pollutants in the region and can contribute to improving or degrading air quality.

The goal of this study is to quantify the relative contribution of the pollutants associated with biomass burning from central

and southern Africa on the surface concentrations of aerosols, carbon monoxide (CO) and ozone ($O_3$) in urbanized areas pertaining to the DACCIWA project. In order to take into account all important sources, a large area is modeled, encompassing southern West Africa (Ivory Coast, Ghana, Togo, Benin, Nigeria) and representing all sites of interest for the DACCIWA project. First, the simulations are compared to observations to quantify the model ability to accurately estimate the meteorology and the chemical species concentrations. Second, we assess the relative contribution of vegetation fires by investigating the

difference between two simulations: one with and one without fires emissions, from now on referred to as the FIRE and NoFIRE simulations, respectively. The chemical composition of the aerosols over coastal SWA is also presented.

Section 2 presents the observations locations and Section 3 presents the models. Section 4 analyzes the meteorological situation. Section 5 presents a tracers release experiment. Section 6 presents an analysis of the long-range transport of gas and aerosol species and Section 7 an analysis of gas and aerosol surface concentrations in the cities located in the coastal areas.

Conclusions are finally presented.

## 2 Observations

Data from very different sources were used to conduct this study. They were obtained from space-borne platforms and groud-based stations. Satellites data provide information on the horizontal and vertical distributions as well as the long-range transport: the Infrared Atmospheric Sounding Interferometer (IASI) for vertically integrated values of CO, the Moderate Resolution

Imaging Spectroradiometer (MODIS) for Aerosol Optical Depth (AOD) and the Cloud-Aerosol Lidar with Orthogonal Polarization (CALIOP) with aerosols types classification. Some other measurements are also available for specific locations such as the British Atmospheric Data Center (temperature and precipitation rate), the AERONET photometers network for AOD and the Sahelian Transect for surface Particulate Matter ($PM_{10}$). Details about these measurements data are provided in the next





sections where the data are used. Finally, note that, for chemistry, there is a lack of in-situ surface measurements for this region and during the studied period.

| Station Name | Country | Longitude ($^o$E) | Latitude ($^o$N) |
|---|---|---|---|
| **Observation sites (AERONET)** | | | |
| Ascension | Saint Helena | -14.41 | -7.98 |
| Bambey | Senegal | -16.45 | 14.70 |
| Banizoumbou | Niger | 2.66 | 13.54 |
| CapoVerde | CapoVerde | -22.94 | 16.73 |
| Cinzana | Mali | -5.93 | 13.28 |
| Dakar (M'Bour) | Senegal | -16.96 | 14.39 |
| Ilorin | Nigeria | 4.34 | 8.32 |
| Izana | Tenerife | -16.50 | 28.30 |
| Lope | Gabon | 11.93 | -0.08 |
| Zinder | Niger | 8.98 | 13.75 |
| **Model analysis, no measurements** | | | |
| Guinean Gulf sites | | | |
| Abidjan | Ivory Coast | -4.01 | 5.34 |
| Lagos | Nigeria | 3.38 | 6.45 |
| Tracers experiment | | | |
| trcW | Gabon | 12.0 | -5.0 |
| trcE | Dem. Rep. of Congo | 25.0 | -5.0 |

**Table 1.** *Measurements stations with their names, countries and coordinates (sorted in alphabetical order). The AErosol RObotic NETwork (AERONET) sites provide AOD measurements. The additional sites quoted "Guinean Gulf sites" correspond to location where there is no measurements but where model results are extracted in order to compare the simulations without and with the fires emissions. The "Tracers experiment" sites correspond to locations where tracers are released for a specific model experiment.*

Several stations are used for (i) the model validation, (ii) the discussion on the biomass burning impact on the surface concentrations. These stations are listed in Table 1 and their location is presented in Figure 1.

# 3  Modelling

For the simulations performed in this study, two regional models are used: (i) the Weather and Research Forecasting (WRF) model calculates the meteorological variables, (ii) the CHIMERE chemistry-transport model calculates the concentrations of the tracers and the gaseous and aerosols species. WRF first calculates meteorological fields. Second, CHIMERE reads the meteorology from WRF and surface emissions to simulate the chemical concentrations. WRF and CHIMERE use the same





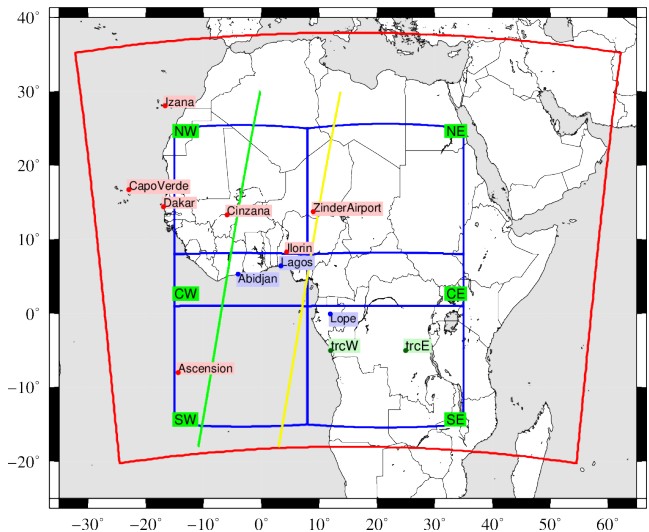

**Figure 1.** *Map of the modelled domain (the red frame). The circles and the locations names indicates the stations described in Table 1: the "red" symbols represent the AERONET stations and the "blue" symbols represent locations representative of the most studied sites in the DACCIWA project. The two lines represent the CALIOP trajectories, with the green one for the 26 July 2014 and the yellow one for the 27 July 2014. The sub-domains defined for the comparisons between the model and the IASI data are in blue.*

horizontal domain and the same grid size of 60 km × 60 km. The modeled period ranges from 1 May to 31 July 2014. The domain size is presented in red in Figure 1.

### 3.1 The WRF meteorological model

The meteorological variables are modeled with the non-hydrostatic WRF regional model in its version 3.6.1, (Skamarock et al.,

2007). The global meteorological analyses from the National Centers for Environmental Prediction (NCEP) with the Global Forecast System (GFS) products are used to nudge WRF hourly for pressure, temperature, humidity and wind. In order to preserve both large-scale circulations and small scale gradients and variability, the 'spectral nudging' technique was applied. This nudging was evaluated in regional models, as presented in Von Storch et al. (2000). In this study, the spectral nudging was selected to be applied for all wavelengths greater than ≈2000km (wave numbers less than 3 in latitude and longitude,

for wind, temperature and humidity and only above 850 hPa). This configuration allows the regional model to create its own dynamics, thermodynamics and composition features within the boundary layer and insures that the large scale follows the analysed thermodynamics fields.

The model is used with 28 vertical levels from the surface to 50 hPa. The Single Moment-5 class microphysics scheme is used, allowing for mixed phase processes and super cooled water, (Hong et al., 2004). The radiation scheme is RRTMG scheme

with the MCICA method of random cloud overlap, (Mlawer et al., 1997). The surface layer scheme is based on Monin-Obukhov with Carlson-Boland viscous sub-layer. The surface physics is calculated using the Noah Land Surface Model scheme with



four soil temperature and moisture layers, (Chen and Dudhia, 2001). The planetary boundary layer physics is processed using the Yonsei University scheme, (Hong et al., 2006) and the cumulus parameterization uses the ensemble scheme of Grell and Dévényi (2002). The aerosol direct effect is taken into account using the Tegen et al. (1997) climatology.

## 3.2 The CHIMERE chemistry-transport model

CHIMERE is a chemistry-transport model allowing the simulation of concentrations fields of gaseous and aerosols species at a regional scale. It is an off-line model, driven by pre-calculated meteorological fields. In this study, the version fully described in Menut et al. (2013a) and updated in Mailler et al. (2017) is used. If the simulation is performed with the same horizontal domain, the 28 vertical levels of the WRF simulations are projected onto 20 levels from the surface up to 200 hPa for CHIMERE. The CHIMERE vertical levels increase in depth from the surface to the top. The altitude (above ground level) of the first four vertical layers are $\approx$ 18 m, 42 m, 75 m, 115 m, respectively. Being expressed in $\sigma$-pressure coordinates, the layer depths are not constant in space and time, and are able to follow the surface pressure evolution as well as the topography.

The chemical evolution of gaseous species is calculated using the MELCHIOR2 scheme. The photolysis rates are explicitly calculated using the FastJX radiation module (version 7.0b), (Wild et al., 2000; Bian et al., 2002). The aerosols are modeled using the scheme developed by Bessagnet et al. (2004). The aerosol size is represented using ten bins, from 40 nm to 40 $\mu$m, in mean mass median diameter (MMMD). The aerosol life cycle is completely represented with nucleation of sulfuric acid, coagulation, absorption, wet and dry deposition and scavenging. The scavenging is represented by in-cloud and sub-cloud scavenging.

The aerosol model species and their characteristics consist in ten different types of aerosols, some being a compound of several aerosol species. In the results section, these species are represented as: PPM is for anthropogenic Primary Particulate Matter, DUST is for mineral dust, EC is for Elemental Carbon, POM is for Primary Organic Matter, SALT is for Sea salts, SOA is for Secondary Organic Aerosols. $SO_4$, $NO_3$ and $NH_4$ are equivalent of Sulfate, Nitrate and Ammonium, respectively. WATER is for water. More details are provided in Menut et al. (2013a) and Menut et al. (2016).

The modeled AOD is calculated by FastJX for several wavelengths over the whole atmospheric column, as detailed in Menut et al. (2016). At the boundaries of the domain, climatologies from global model simulations are used. In this study, outputs from LMDz-INCA (Hauglustaine et al., 2014) are used for all gaseous and aerosols species, except for mineral dust where the simulations from the GOCART model are used (Ginoux et al., 2001).

The anthropogenic emissions are issued from the "Hemispheric Transport of Air Pollution" (HTAP) global database, (Janssens-Maenhout et al., 2015). These emissions are provided as monthly databases. For the simulation, weekly profiles are applied to have week-day, Saturday and Sunday. In addition, hourly profiles are applied to have an hourly variability also depending on the activity sector. The complete calculation of these fluxes is detailed in Menut et al. (2012); Mailler et al. (2017). An example for $NO_2$, representative of anthropogenic emissions, (g/m$^2$/day) is presented in Figure 2 for a week-day. In the Guinea Gulf, these emissions are of the same order of magnitude as in the south of Europe and correspond to the megacities located between Abidjan and Lagos.



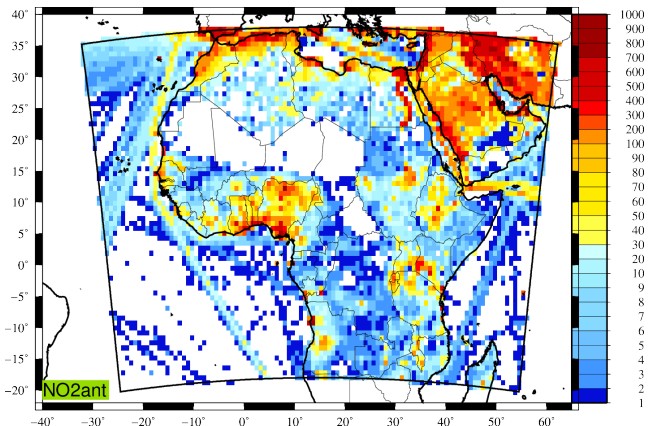

**Figure 2.** *Anthropogenic emissions surface fluxes of $NO_2$ ($g/km^2/day$) for a week day of July 2014 and over the simulation domain.*

### 3.3 The mineral dust emissions

The mineral dust emissions are calculated using the Alfaro and Gomes (2001) scheme, optimized following Menut et al. (2005)
and using the soil and surface databases presented in Menut et al. (2013b). Since this latter article, several changes were done
in the emissions scheme. They are all related to the spatial extent of the emissions fluxes calculations: from the Sahara only
to any arid or semi-arid areas in the world. The surface and soil databases being global, the fluxes are now systematically
calculated over the whole domain, including non-desert areas, such as Europe. In order to keep realistic fluxes under a variety
of meteorological conditions, the emissions scheme was adapted. These changes are active for all model cells including the
desert ones. These changes are briefly described below:

The erodibility is diagnosed using the United States Geological Survey (USGS) land use and an additional database, built
using MODIS surface reflectance, (Beegum et al., 2016). For all model cells considered as 'desert', the MODIS erodibility
is used while for all other cells, a constant erodibility factor is applied depending on the USGS land use, as in (Menut et al.,
2013b). To take into account the rain effect on mineral dust emissions limitation, a 'memory' function is added. During a
precipitation event, the surface emissions fluxes are set to zero. After the precipitation event, a smooth function is applied to
account for a possible crust at the surface and, thus, fewer emissions, (Mailler et al., 2017).

### 3.4 The biomass burning emissions

The fires in Central Africa generally start in April and peak in July, (Cooke et al., 1996; Barbosa et al., 1999). A lot of
parameters are involved in the calculation of these emissions, making the wild-fires fluxes one of the most uncertain source in
chemistry-transport models, (Grell and Baklanov, 2011; Turquety et al., 2014). This flux calculation may be divided into three
parts:





1. The emissions fluxes: this is the value of surface fluxes, depending on the burned area projected on the grid cell surface and for each chemical species taken into account in the chemistry-transport model.

2. The injection height: this parameter defines the top altitude of the fires emissions vertical plume.

3. The injection vertical profile: having the total emitted mass flux and the top of the plume, it is necessary to define the shape of the vertical injection profile.

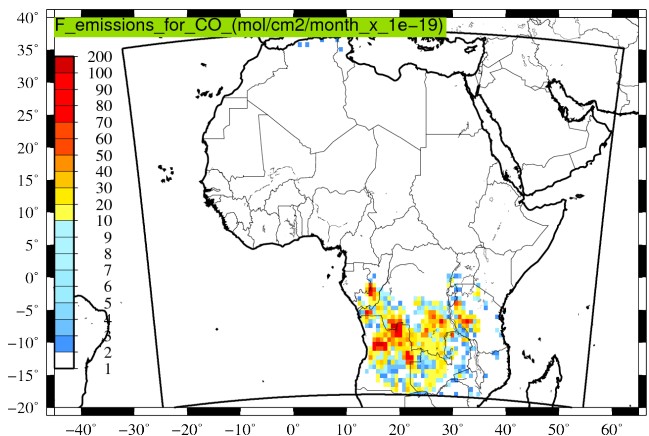

**Figure 3.** *Biomass burning emission fluxes of CO (in molecules/cm$^2$/month) cumulated over the whole month of July 2014.*

The underlined emissions fluxes depend on the burned area, land-use, vegetation type, and fuel load. The calculations are done at an hourly time step using the high-spatial resolution Analysis and Prediction of the Impact of Fires on Air Quality Modeling (APIFLAME) model. All information about this estimation are provided in Turquety et al. (2014). This model was previously used, for example, in Rea et al. (2015). In this APIFLAME model version, fire emissions fluxes are calculated based on the MODIS area burned product MCD64, (Giglio et al., 2010). The fluxes being daily estimated, a diurnal profile is applied where 30% of the daily is redistributed during the night (18:00 to 8:00 LT-local time) and 70% during the day, close to values usually chosen in biomass burning model studies, (Zhang et al., 2012). An example of the time cumulated flux of CO for month of July 2014 is presented in Figure 3. Emissions related to biomass burning are mainly located in Central Africa.

For the injection height, $H_p$, we used the approach proposed by Sofiev et al. (2012). In South-Africa and during the months of July and August, a typical variability of $H_p$ is estimated between 3 and 4.5 km, (Labonne et al., 2007). The calculation of Sofiev et al. (2012) is based on the Convective Available Potential Energy estimation, itself diagnosed using the Fire Radiative Power (FRP) of each fire. They validated their $H_p$ calculation using the Multi-angle Imaging SpectroRadiometer plume height retrievals and showed a good agreement between the two. $H_p$ is estimated, for each individual fire, as:

$$H_p = \alpha H_{abl} + \beta \left( \frac{P_f}{P_{f0}} \right)^{\gamma} exp \left( -\frac{\delta N_{FT}^2}{N_0^2} \right) \tag{1}$$



with $\alpha$=0.24, $\beta$=170 m, $\gamma$=0.35, $\delta$=0.6, $P_{f0}$=$10^6$ W and $N_0^2$=$2.4 \times 10^{-4}$ s$^{-2}$. The FRP, $P_f$, is expressed in W (with 1 W = 1 J s$^{-1}$ = 1 m$^2$ kg s$^{-3}$). $N_{FT}$ is the Brünt-Vaisala frequency in the free troposphere.

An empirical correction is performed for the known underestimation of FRP by MODIS in case of strong fires, (Veira et al., 2015):

$$P_f^* = P_f \times \left( \frac{H_p}{H_{deep}} \right)^\epsilon \qquad (2)$$

with $\epsilon$=0.5 and $H_{deep}$=1500 m.

The shape of the injection vertical profile is difficult to estimate. But this is a very sensitive parameter: for a same amount of emissions, the way to vertically distribute this amount will completely change the long-range vertical transport. As, in this study, the goal is to estimate if the biomass burning emissions can reach the surface of the north coast of the Guinean gulf, this vertical injection needs a particular attention.

A lot of global models are simply injecting the emitted mass in an homogeneous way in the boundary layer or from surface to a prescribed $H_p$ (see references in Sofiev et al. (2012), among others). Sofiev et al. (2013) distribute the flux homogeneously between $H_p/3$ and $H_p$. Other models use more complex parameterizations as thermal convective approaches, primarily developed for boundary-layer convection in dynamical models and adapted to the specific problematic of pyroconvection, (Freitas et al., 2007; Rio et al., 2010). But, this 'thermal' approach is numerical cost consuming and difficult to use, being very sensitive to the chosen input parameters. Finally, some vertical profiles are close to the vertical diffusivity profile, $K_z$, shape with the maximum of injection in the middle of the $H_p$ height such as in Raffuse et al. (2012); Veira et al. (2015).

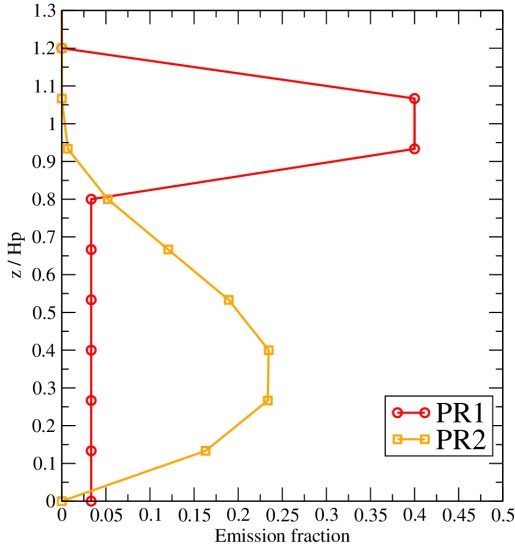

**Figure 4.** *Vertical profiles of factors used for the injection of biomass burning emissions in the troposphere.*





In this study, and in order to reduce the uncertainty of our results, two simulations are performed with different injection vertical profiles (all other model parameters remain identical). The two profiles are displayed in Figure 4:

–    PR1: 80% of emissions are injected in the model layers included in the interval $0.9 \times H_p < z < 1.1 \times H_p$. The rest, 20%, is injected between the surface and $0.9 \times H_p$. This profile was selected to: (i) estimate the long-range transport of fires, (ii) see if fires mainly injected in the mid-troposphere may have an impact on remote surface concentrations. This profile

represents an idealized shape of what usually diagnose 'thermal' parameterizations under convective periods.

–    PR2: The emissions are injected between the surface and $H_p$. The $H_p$ value is estimated for each fires. This profile shape is close to the ones used in Veira et al. (2015). This profile is realistic and the simulations done with this profile will be later used for the results discussion. This profile has a $K_z$-like shape and is thus expressed as:

$$\begin{cases} \text{if } z_n \leq 1 & EF(z) = H_p\, z_n\,(1 - z_n)^2 \\ \text{if } z_n > 1 & EF(z) = 0 \end{cases} \tag{3}$$

with $z_n = z\,/\,H_p$.

### 3.5 Simulations

Four different simulations are performed over the whole period, from 1st May to 31 July 2014:

•    TRC: these simulations consist in tracer release experiments. They are dedicated to answer the question: *what are the regions of Central Africa for which biomass burning can reach the coastal cities of the Guinean Gulf?* The tracers are released from

two locations: in the western and eastern part of the biomass burning area in Central Africa (Table 1). The corresponding experiments are named 'trcW' and 'trcE'. For each location, two injection heights are used: experiments for which aerosols are injected between the surface to 3000 m AGL are labelled '1' and experiments for which aerosols are injected from 3000 m to 6000 m AGL are tagged '2'. These two altitudes intervals enable to estimate the sensitivity of the biomass burning transport to different regimes of $H_p$ values. The tracers are defined as aerosols, with a density and a size distribution, and

are thus subject to deposition during transport. The tracers are continuously released from 15 June to 30 July. There is no diurnal cycle, the emissions flux being constant during the whole period. The released amount is arbitrary and has no unit (but for realism, the emitted fluxes are of the same order of magnitude than anthropogenic emissions).

•    NoFIRE: this simulation takes into account all processes (dynamic and chemistry) available in the CHIMERE model. All emissions are taken into account except the biomass burning emissions. This simulation will provide the gas and aerosol

atmospheric content without biomass burning.

•    FIRE PR1 and FIRE PR2: These simulations have the same configuration as the NoFIRE simulation except that we add the vegetation emissions fluxes. These emissions fluxes are injected in the troposphere following the two injection height profiles PR1 and PR2, described in the previous section. By difference with the NoFIRE simulations, we will be able to quantify the impact of the biomass burning on the gas and aerosol atmospheric content.



The simulations with the tracers and with the complete chemistry with fires enable to have also answers about the way to model the biomass burning injection in the troposphere. The TRC simulation is dedicated to estimate the influence of two major vertical ranges. This is relevant to estimate the impact of different $H_p$ injection heights (even if the values are fixed to better undertand the different pollutants transport pathways). The FIREs simulations use a parameterized $H_p$. In this case, and with the PR1 and PR2 simulations, we study the sensitivity of the results to the shape of the injection height profile, considering that

the modelled $H_p$ value is the best as possible estimate.

In order to analyze the modelled results, complementary informations are provided in Figure 1: (i) the modelled domain (in red), (ii) the two CALIOP trajectories (the green one for 26 July 2014 and the yellow one for 27 July 2014), (iii) the location of the two tracers release hotspots and (iv) the six sub-domains defined for the model versus IASI data comparisons. These six sub-domains, in blue, are defined to represent several regions as:

• SW: Bottom-West is the only sub-domain completely over the sea, and may be under the plume of biomass burning coming from Central Africa.

• SE: Bottom-East represents the region in Central Africa where vegetation fires are observed.

• CW: Central-West is the sub-domain containing the Guinean Gulf cities studied in this article.

• CE: Central-East may be under the plume of vegetation fires coming from South-East.

• NW and NE: North-West and North-East correspond to regions without vegetation fires emissions but with mineral dust emissions.

## 4   Synoptic meteorological situation

The studied period corresponds to a specific and complex meteorology. In this section, we focus on precipitation near the coastline, where various precipitating systems occurred during the period from May to August, Figure 5. This constrains the

transport of local emissions as well as impact the wet deposition of emitted species.

Time-series of comparisons are presented in Figure 6 for the highly urbanized coastal cities of Lagos and Abidjan. The observations are extracted from the Centre for Environmental Data Analysis (on the BADC database *http://badc.nerc.ac.uk/home*) and correspond to the Met Office MIDAS Land Surface Stations data (*http://data.ceda.ac.uk/badc/ukmo-midas/*). They are provided with a 3-hourly time step and are daily accumulated. In Lagos, the observed precipitation rate is sporadic but in-

tense, with values up to 60 mm/day five times during the period. For May and June, the model simulates lower values for these events. During July, the model simulates the two largest precipitation events on 2 and 18 July, but with a time shift of 1 to 2 days, respectively. Furthermore, the model produces rain every day, unlike what is observed, thereby overestimating the number of rainy days. This will likely lead to an underestimation of the modeled surface concentrations, due to the enhanced simulated wet scavenging in the lower troposphere. In Abidjan, the observed precipitation rate is more important and frequent. The simulation is more realistic and there is a better agreement between the number of rainy days and the 24-h accumulated

5    precipitation. The two rainiest periods, around 15 June and 1st July, are well simulated, with rainfall amounts in excess of 50 mm/day.



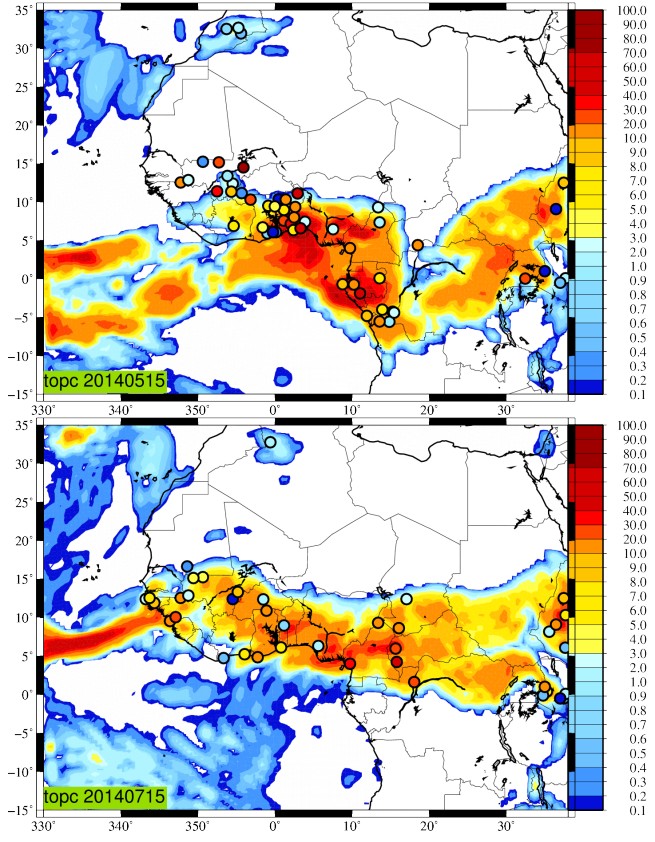

**Figure 5.** *Comparison between observed (BADC) and modeled daily cumulated precipitation rate (mm/day) for the 15 May and 15 July 2014. For the precipitation measurements, only the non-zero daily cumulated values are reported on the plot.*

## 5 The tracers experiment

### 5.1 Time series of surface concentrations

For the TRC simulations, the results are presented in Figure 7 for the four emitted tracers and for three sites, Lope, Lagos and
Abidjan. The goal is to estimate if the emissions which occurred in Central Africa may reach these sites.

The first result is that the four emitted tracers provide non-zero surface concentrations on the three sites. It means that the
meteorological conditions are favourable to transport biomass burning emissions from Central Africa to the Guinean Gulf
cities.

Lope is close to the most important biomass burning observed during the modelled period. The tracers are first emitted
on 15 June and the first non-zero tracer concentrations in Lope are modelled on 17 June. As expected, the most important
concentrations are modelled for the trcW1 tracer, the site being very close to the source. The values are important (up to 500
in arbitrary unit). For the same source, but emitted in altitude in trcW2, the concentrations are lower but not negligible. This





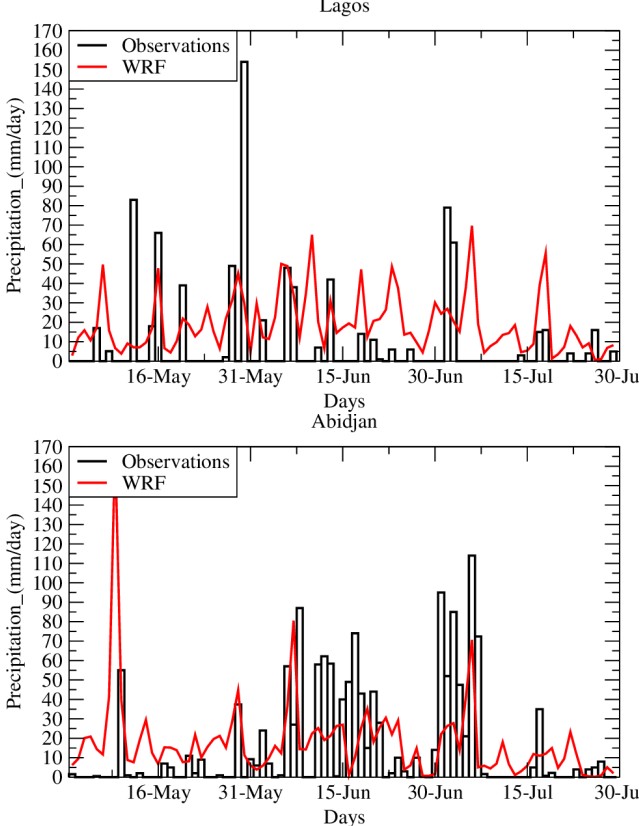

**Figure 6.** *Time series of 24-h accumulated precipitation from the BADC stations and the corresponding model cell: in Lagos (top), and Abidjan (bottom).*

shows that even if a tracer is emitted between 3000 and 6000 m AGL, the daily dry convection the lower troposphere is strong enough to mix non negligible concentrations down to the surface layer. The tracers experiments further east (trcE1 and trcE2) have also non negligible concentrations in the surface layer in Lope. The first non-zero values are modelled on 23 June, 8 days after the initial tracer emissions. This means that, even, if the emissions are far to the east, the mixing and long-range transport bring aerosols to the coast in one week.

Even if Lagos and Abidjan are far from the tracers sources ($\approx 1000$ km), there is an impact of the biomass burning on their surface concentrations. The most important concentrations are modelled for the tracer emissions in the western domain. For this location, the peak values are not completely correlated in time, depending on the altitude of injection. This shows than the main biomass burning plume follows the same transport in the troposphere, but also that mixing coupled with differential advection may change the final trajectories in the surface layer before reaching the studied cities. Finally, note that in Abidjan, the highest impact is due to the "altitude tracer" trcW2 and not to the "surface" tracer trcW1. The concentrations for the east





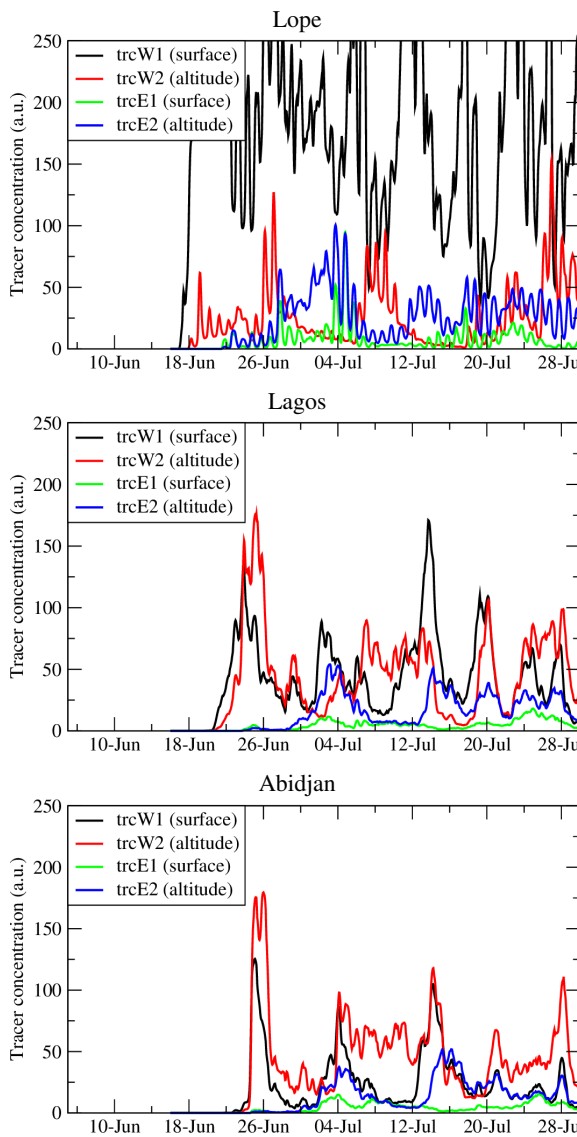

**Figure 7.** *Time-series of surface concentrations (arbitrary units) in Lope, Lagos and Abidjan for the four tracers release from 15th June to 31 July and close to the most important biomass burning emissions observed in Central Africa.*

5   tracer are 4 to 5 times lower than the west tracer, showing that some concentrations are arriving in the studied sites but the main part is probably transported to other places, far from the Guinea gulf northern coast.



## 5.2 Maps of tracers surface concentrations

To increase our understanding of the complex transport pathways of the several studied aerosols, we focus on the 27th July 2014. This day being at the end of the studied period, it corresponds to the maximum of potential long-range transport from 10 Central Africa to the Guinean Gulf cities.

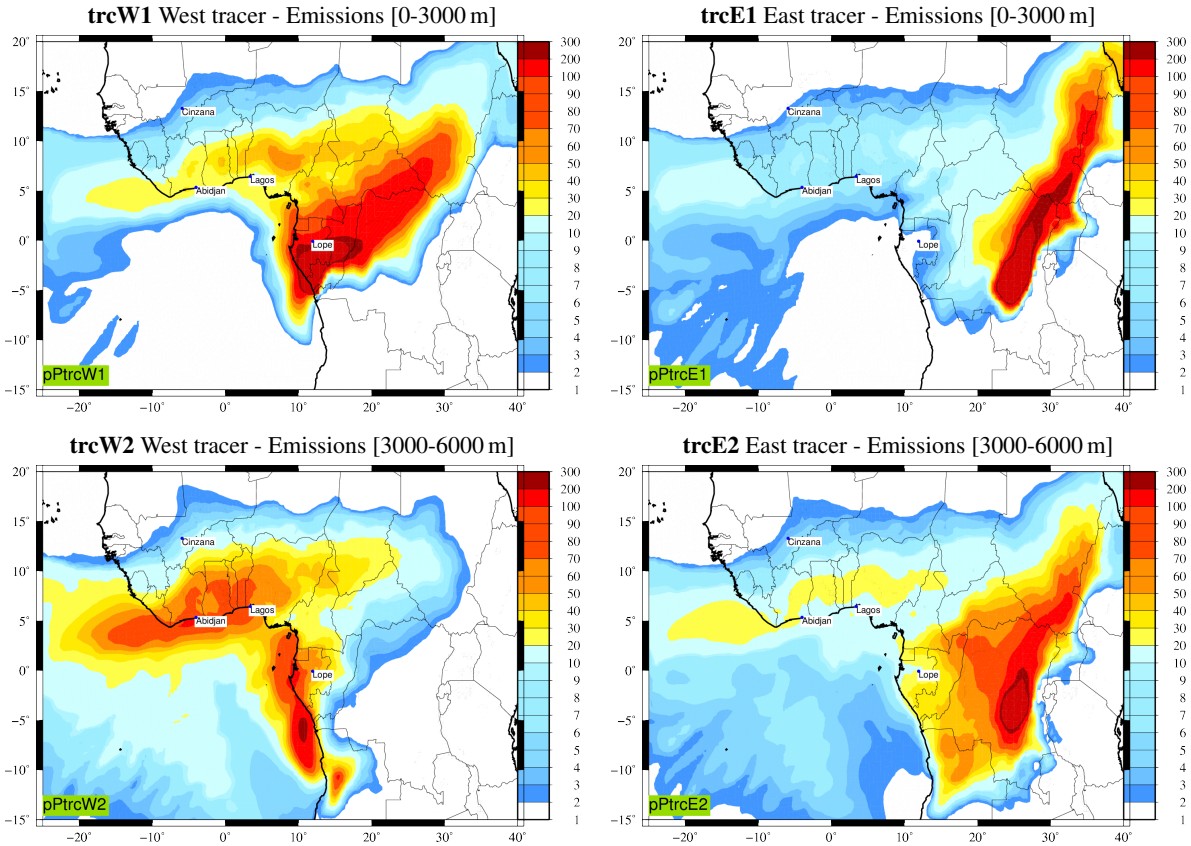

**Figure 8.** *Maps of tracers surface concentrations (arbitrary units) for the 27 July 2014 at 12:00 UTC. Each map refers to a tracer with trcW1 (west surface), trcW2 (west altitude), trcE1 (east surface) and trcE2 (east altitude).*

Figure 8 presents surface concentrations for the 27 July 2014 at 1200 UTC and for each of the four tracers. As previously seen, the four tracers reach the Guinean Gulf cities of Lagos and Abidjan. The most important transport from the fire region to these cities is with the western tracer trcW. For trcW1, the main transport from the emission is going to the south and the north-east. Up to the latitude $\phi = +5^oN$, the tracer transport changes and follows the Harmattan to the west. The most important contribution comes from the tracer emitted in altitude, trcW2. A large part is observed in the southern part of the emission, when another contribution follows the coastline, then, as for trcW1, follows the main flow from east to west, to finally arrive 5 in the Lagos and Abidjan.



For trcW2, the main part of the concentrations is transported to the east. But, even for a longitude $\lambda > 20^o$N, a part of this plume is also caught by the main transport of the Harmattan and continues to the west. Less important than for trcW1, a non-negligible part of trcW2 is observed in Lagos and Abidjan.

This tracer experiment allows to better understand the complex transport pathways of the biomass burning aerosols from Central Africa to the cities of Lagos and Abidjan. This can be summarized as follows:

- Over continent Central Africa, the main transport pathway for biomass burning aerosols is towards the north-east. For fires, in the western part of the emissions region, the aerosol plume may follow the coastline.

- The biomass burning products, mainly occurring during the day, are rapidly mixed in the boundary layer. This boundary layer is very deep and may reach 3000 to 4000 m AGL. This means that a few hours after the emissions, this is a vertical constant profile which is advected.

- The part of the plume going to the west is already vertically well-mixed when it passes from land to sea. A part is thus transported in the marine layer, another part up to the marine layer, in a well stratified layer in the free troposphere.

- Whatever the emissions location and the injection height, the plume is changing direction when arriving up to latitude $\phi = +5^o$N: it is then transported to the south-west, following the Harmattan flow.

## 5.3 Summary of results

The first conclusion for this part is that the whole area of biomass burning in Central Africa is impacting the surface concentrations in the Guinean Gulf coast cities. Second, the transport pathways are different if the fires are emitted in the lower or in the mid troposphere. But, in the end, after a few weeks, the fires emissions occurring in the mid-troposphere have an impact of the same order of magnitude than those emitted in the boundary layer. For the specific case of 27th July 2014, the tracers emitted at the surface are flowing to the north-east, specifically those emitted to the east of the fires area. More in altitude, and for the western part of the fires emissions, the tracers are flowing along the coast and are caught in the Harmattan flux, then vertically mixed, to finally reach the Guinean Gulf cities.

## 6 Long-range transport of gas and aerosol species

Before analyzing local pollution, it is necessary to have a synoptic view of the long-range transport of pollutants. In the previous section, it was shown that the meteorological conditions are favourable to import Central Africa pollutants to the Guinean Gulf coast. In this section, and using available data, the simulations with realistic emissions, transport and chemistry are used in order to quantify the model ability to retrieve the main pollutants variability and intensity. The discussion is organized based on surface $PM_{10}$ measurements, satellite retrievals of CO columns from IASI and of AOD from MODIS and AERONET. For each dataset, a comparison is presented with the two simulations using PR1 and PR2.



## 6.1   AOD CHIMERE Vs MODIS

5   The MODIS AOD product at $\lambda$=550nm (from the MODIS/Terra Aerosol 5-Min L2 Swath 10 km data collection 5.2) is used to quantify the increase of aerosol due to biomass burning (Levy et al., 2010). The model outputs and observations are collocated in space and time in order to compare exactly the model to the available observations.

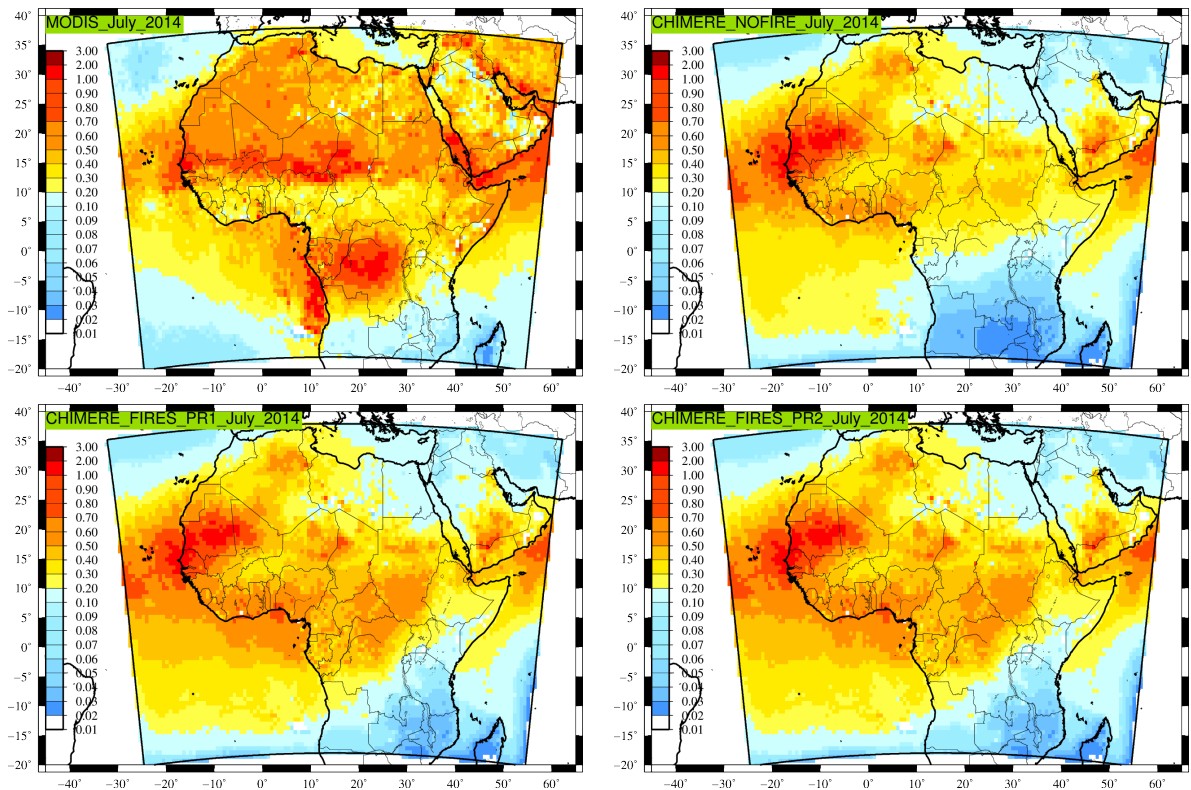

**Figure 9.** *Monthly averaged horizontal distribution of AOD ($\lambda$=550nm) for MODIS, CHIMERE without fires and CHIMERE with vegetation fires emissions and the two fires injection height profiles, PR1 and PR2.*

Results are presented in Figure 9 for the month of July 2014, when biomass burning is at its maximum in intensity for the studied period. The satellite data are compared to three model configurations: (i) without fires with the NoFIRE simulation, (ii)

10   with fires and for the injection heights PR1 and PR2.

Over Africa, the MODIS data show two large areas of AOD>0.5: in Central Africa (corresponding to fires emissions) and up to the latitude $\phi$=10$^o$N (corresponding to mineral dust emissions). Without fires emissions, the NoFIRE simulation enables to validate the mineral dust modelling and shows the model tends to underestimate the AOD between 10 and 15 $^o$N. On the other hand, the plume transported to the Atlantic is slightly over-estimated. Over the Guinean Gulf, the modelled AOD are overestimated (0.5 when MODIS shows 0.3). The AOD due to mineral dust is mainly under-estimated and many factors may explain this. As already discussed in Menut et al. (2016), the modelled size distribution may be inaccurate while it is very



sensitive for the AOD estimation. It was also shown that a bias in AOD calculation may exist but is not necessarily related to erroneous modelled surface concentrations of Particulate Matter (PM). Over this region and during this period, an additional explanation of this bias could be related to the way the model handles the precipitation events. The results presented in section 4 showed that the modelled precipitation patterns correspond to what was observed with the BADC stations. But, as discussed in Ruti et al. (2011); Flaounas et al. (2011); Efstathiou et al. (2013), these processes remain highly variable, uncertain and difficult to validate and this is possible that the scavenging was not modelled enough correctly, leading to these differences between model and observations.

When including the calculation of biomass burning emissions and their transport, a general increase is observed in the fires simulations. While AODs are less than 0.05 in the NoFIRE simulation, AOD values can reach 1 over Cameroon in the FIRE simulations. The westerly winds transport these biomass burnings plumes over the Guinean Gulf and the model results shows that the whole coast is under these dense plumes, from Nigeria to Ivory Coast. With MODIS, two high AOD regions related to fires are observed. One in Central Africa and the other along the coast. With the model, the increase of AOD is located more to the north and less intense. Finally, note that there are no significant differences between the results of the two PR1 and PR2 simulations.

The conclusion for this part is that the model reproduces the two large areas of high AOD, due to mineral dust and biomass burning emissions. But the intensities are not well retrieved: over the Central Africa, the modelled AODs due to biomass burning are under-estimated. This may be due to fires intensity or size distribution of the modelled aerosol. This will be discussed in the next section 6.3 with the comparison between observed and modelled CO.

## 6.2 AOD and Angstrom coefficient CHIMERE Vs AERONET

The aerosol optical properties are compared between observations and model using the AERONET measurements (Holben et al., 2001). The comparison is done using (i) the AOD measured by the AERONET photometers and for a wavelength of $\lambda$=550 nm and using the level 2 data, (ii) the Angström coefficient calculated using the AOD measured for $\lambda$= 470 and 870 nm. The comparisons for the simulations NoFIRE and FIREs are presented for the stations listed in Table 1.

Results are presented as statistical scores in Table 2 and as time series in Figure 10 (left column) for the AOD. The scores are calculated with an hourly time-step. The percentage of valid available data is provided in Table 2. Scores are only presented for a given AERONET station if data is acquired on a regular basis over a period of three months (i.e. 2280 hours) and if more than 30 values can be used to compute them (only scores where at least 1.5% of data are available are shown). For the time series, the two simulations with fires PR1 and PR2 are displayed. But, for the scores, only the results for PR2 are presented, the differences between PR1 and PR2 being negligible. Note that, except for Lope, most of the AERONET stations are located in the northern part of the studied region, mainly under the influence of mineral dust emissions.

In Table 2, and except for the Lope station, differences between the simulations NoFIRE and FIREs are very small. The correlations values range between -0.08 (Ascension) and 0.77 (Lope). The low score in Ascension is related to the location of the site and the fact that the long range transport over the sea is difficult to reproduce. For the other sites, the correlations are larger and show that the mineral dust variability is well modelled. The only site with differences between NoFIRE and FIREs





| Site | $F$ | $N$ | Obs | Model | $R_t$ | Bias |
|---|---|---|---|---|---|---|
| Ascension | 0 | 24.3 | 0.09 | 0.26 | -0.08 | 0.17 |
|  | 1 | 24.3 | 0.09 | 0.27 | -0.06 | 0.18 |
| Banizoumbou | 0 | 2.0 | 0.30 | 0.25 | -0.32 | -0.06 |
|  | 1 | 2.0 | 0.30 | 0.27 | -0.46 | -0.03 |
| CapoVerde | 0 | 15.8 | 0.43 | 0.52 | 0.56 | 0.09 |
|  | 1 | 15.8 | 0.43 | 0.52 | 0.56 | 0.09 |
| Cinzana | 0 | 30.2 | 0.52 | 0.43 | 0.39 | -0.09 |
|  | 1 | 30.2 | 0.52 | 0.44 | 0.39 | -0.08 |
| Dakar | 0 | 38.7 | 0.56 | 0.57 | 0.69 | 0.01 |
|  | 1 | 38.7 | 0.56 | 0.58 | 0.69 | 0.01 |
| Ilorin | 0 | 8.4 | 0.35 | 0.44 | 0.39 | 0.09 |
|  | 1 | 8.4 | 0.35 | 0.48 | 0.28 | 0.13 |
| Izana | 0 | 51.4 | 0.04 | 0.19 | 0.59 | 0.14 |
|  | 1 | 51.4 | 0.04 | 0.19 | 0.59 | 0.15 |
| Lope | 0 | 2.8 | 0.34 | 0.15 | 0.46 | -0.19 |
|  | 1 | 2.8 | 0.34 | 0.21 | 0.77 | -0.13 |
| Zinder | 0 | 34.7 | 0.59 | 0.62 | 0.42 | 0.03 |
|  | 1 | 34.7 | 0.59 | 0.63 | 0.41 | 0.04 |

**Table 2.** *Correlations between observations (AERONET) and model (CHIMERE PR2) for the Aerosol Optical Depth (AOD). F is 0 for the NoFIRE simulation and is 1 for the simulation with fires emissions. N is the percentage of hourly available measurements, $R_t$ is the temporal correlation, the bias is calculated by the difference (model minus observation).*

5    is Lope, close to the biomass burning areas. The correlation increases from 0.46 to 0.77 when adding the fires. This shows that the timing of the fires emissions as well as the transport are precise enough to clearly improve the simulation.

Examples of detailed comparisons between AERONET and the model are displayed in Figure 10. In Cinzana, the AOD hourly variability is well retrieved and the majority of observed AOD peaks are modelled. The site being mainly under the influence of mineral dust emissions, there is no significant difference between NoFIRE and FIREs. This is very different in

10   Lope. The addition of the fire emissions increases the AOD during the whole period. The modelled AOD remains lower than the observations, but the timing and the absolute value are more realistic.

As opposed to the comparison with MODIS, these time series and correlation values show that the AOD is not always overestimated by the model. This results shows the large variability obtained with different sets of data, and, also, reflect the difficulty to model this parameter, strongly dependent on the hypothesis done on the optical properties of the modelled aerosols, as well as the way to estimate the extinction with the modelled size distribution (in our configuration, ten bins may be considered as a correctly resolved size distribution for a CTM).



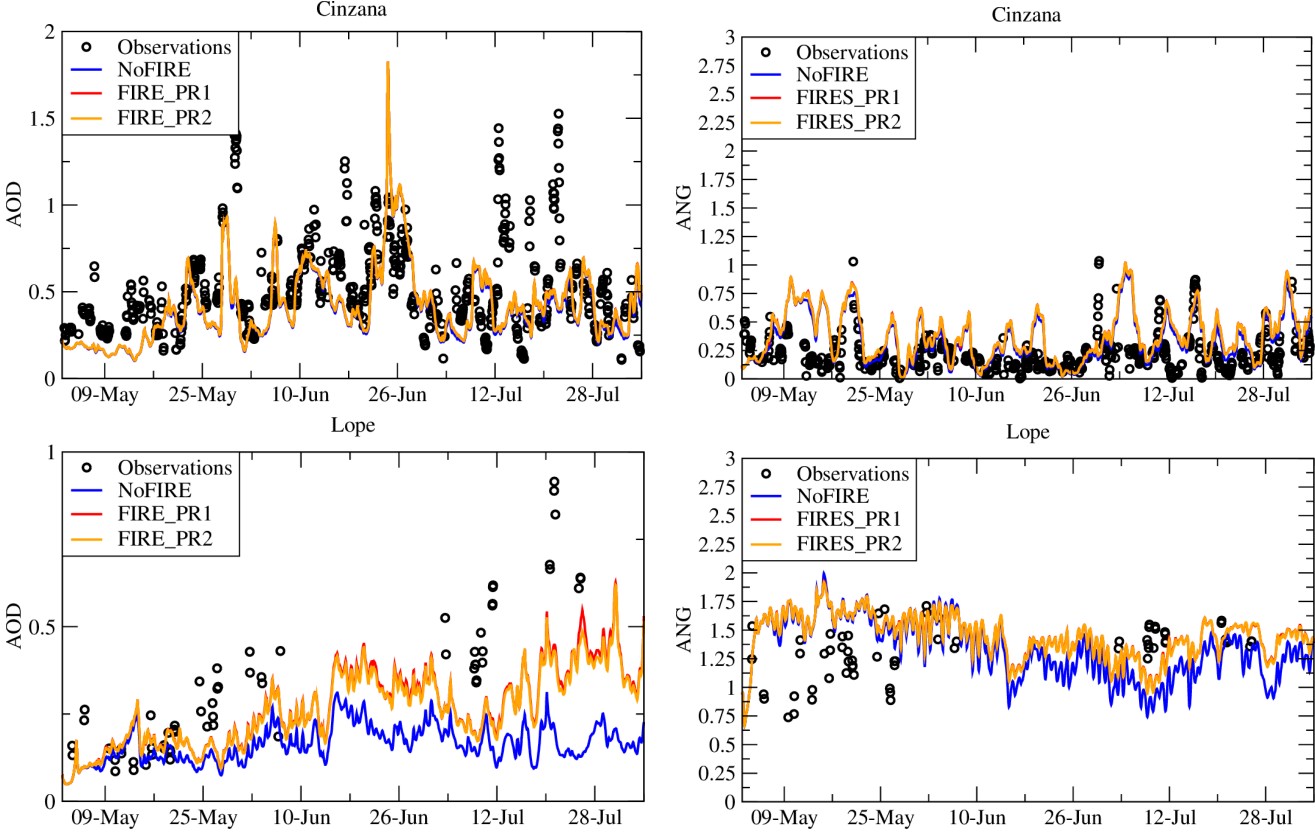

**Figure 10.** *Time series of Aerosol Optical Depth for the two simulations (NoFIRE and FIRE) and the AERONET measurements at the Cinzana and Lope stations.*

Complementary to the AOD, the Angström exponent is also compared to the AERONET retrievals and for the same two stations of Cinzana and Lope. Results are presented in Figure 10 (right column). This exponent expressed the ratio between the AODs at two different wavelengths and its value is inversely proportional to the aerosol size. Low values of the Angström exponent will be representative of mineral dust (aerosols mainly in the coarse mode) when high values will be representative of biomass burning. In Cinzana, the Angström exponent is low, with values between 0 and 0.3 (except some peaks). This means that the aerosol content is mainly mineral dust. On the other hand, in Lope, the Angström exponent is higher and values range between 1 and 1.75, representative of finest particules and, thus, to concentrations related to biomass burning emissions.

## 6.3 CO CHIMERE Vs IASI

The IASI CO total columns retrievals by the FORLI algorithm (Hurtmans et al., 2012; George et al., 2009; Clerbaux et al., 2009) are used. CO is a product of incomplete combustion and, with a lifetime of several weeks. It can be used here as a

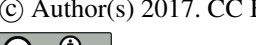



tracer of biomass burning long-range transport. These observations are thus used to check if the biomass burning is accurately

5 spatially calculated and if the main plumes are correctly transported.

**Figure 11.** *Time series of vertically integrated carbon monoxyde (CO column) in $10^{18}$ molecules/cm$^2$ for IASI, CHIMERE without (NoFIRE) and with biomass burning (FIREs PR1 and PR2).*

The comparison between the model and the IASI observations consists in three-days averaged column integrated CO concentrations. The model outputs are collocated in space and time with the satellite observations when they are available. They are also vertically corrected using the satellite averaging kernels before the vertical integration. The comparison is presented



in Figure 11 as time series with the daytime IASI measurements and the corresponding model results. For the model, three simulations are presented: NoFIRE and the PR1 and PR2 fires simulations. Each time series correspond to the sub-domain described in section 2. As a preliminary result, and as shown with the previous results, there is no significant difference between the PR1 and PR2 simulations.

The IASI data show the increase of CO concentrations over Central Africa and the Eastern Atlantic, from May to July (sub-domains SW and SE): under the influence of biomass burning emissions, the CO concentrations are increased by 100%, from $\approx 1.5 \ 10^{18}$ to $\approx 3 \ 10^{18}$ molecules cm$^{-3}$.

For the NoFIRE simulation, the CO concentrations are quasi-constant. For the simulations with fires, the observed CO increase is correctly reproduced. If this increase is slightly under-estimated by the model in the southern part (SW and SE), the time variability and intensity is better modelled in the central part (CW and CE), where are located the studied cities. North of the studied region (NW and NE), the biomass burning emissions has a very low impact on the CO concentrations, the outputs of the simulations NoFIRE and FIRES being close. At this latitude, the model tends to slightly underestimate CO concentrations (by $\approx 0.2 \ 10^{18}$ molecules cm$^{-3}$) with respect to IASI.

In conclusion, it was shown that the CO increase due to biomass burning is observed in areas covering the Guinean Gulf and the studied coastal cities. For these regions, the addition of biomass burning emissions enables to simulate the CO increase observed with IASI during the period. The absolute values and the time variability of CO are well reproduced by the model and are close to the observations.

## 6.4 PM$_{10}$ CHIMERE Vs surface measurements

A complementary comparison to data is performed with the surface PM$_{10}$ measurements of Sahelian Dust Transect (SDT), (Marticorena et al., 2010). With this comparison, we want to ensure that the aerosol mass is well modelled close to the surface. SDT is a network of four stations: Banizoumbou (Niger), Cinzana (Mali), M'Bour and Bambey (Senegal). These stations are colocated with the AERONET stations. The main goal of this network is to have measurements along an iso-latitude transect at $\phi \approx 13 \ ^oN$. In the framework of observations/modeling studies, these measurements were already used in Hourdin et al. (2015), for example.

Statistical scores are presented in Table 3 and for the PR2 configuration only (no difference being found between PR1 and PR2). Results show that the addition of fires emissions has a very low impact on these surface concentrations. This is mainly due to the fact that the only sites having PM$_{10}$ surface concentrations measurements are located in the northern part of the domain and are not under the effect of biomass burning emissions, but mostly under mineral dust emissions and transported plumes. This confirms that the fires plumes do not reach this latitude of $\phi=13^oN$.

An example of time series is presented in Figure 12 for the site of Cinzana. Results also show that the PM$_{10}$ concentrations have a large temporal variability, both in measurements and model. However, even if the correlations are low, it is shown that the model is still able to correctly estimate the amount of mineral dust.




| Site | $F$ | $N$ | Obs | Model | $R_t$ | Bias |
|------|-----|-----|-----|-------|-------|------|
| Bambey | 0 | 99.9 | 74.56 | 73.86 | 0.29 | -0.70 |
|        | 1 | 99.9 | 74.56 | 73.98 | 0.29 | -0.57 |
| Banizoumbou | 0 | 98.8 | 194.70 | 60.58 | 0.13 | -134.12 |
|             | 1 | 98.8 | 194.70 | 61.55 | 0.13 | -133.16 |
| Dakar | 0 | 99.8 | 71.11 | 84.89 | 0.19 | 13.78 |
|       | 1 | 99.8 | 71.11 | 85.01 | 0.19 | 13.90 |
| Cinzana | 0 | 99.0 | 95.60 | 63.89 | 0.25 | -31.72 |
|         | 1 | 99.0 | 95.60 | 64.67 | 0.25 | -30.93 |

**Table 3.** *Correlations between observations (Sahelian Transect) and model (CHIMERE PR2) for the $PM_{10}$ surface concentrations. F is 0 for the NoFIRE simulation and is 1 for the simulation with fires emissions. N is the percentage of hourly available measurements, $R_t$ is the temporal correlation, RMSE the Root Mean Squared Error and Bias, the bias calculated by the difference between the observation and the model.*

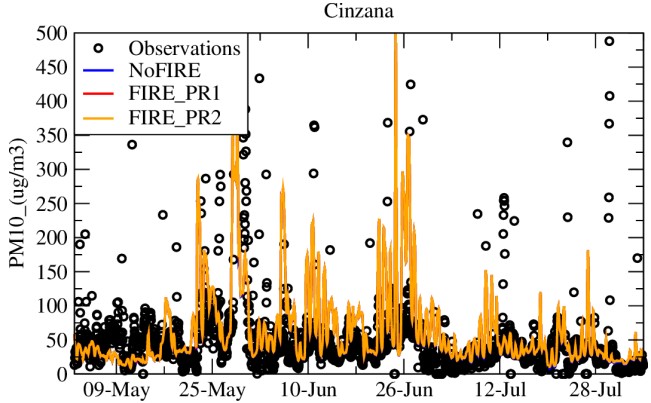

**Figure 12.** *$PM_{10}$ surface concentrations time series measured with the Sahelian Transect Network and modelled with the NoFIRE and the FIREs PR1 and PR2 configurations.*

## 6.5 CHIMERE Vs CALIOP aerosol sub-types

In order to have a synthetic view on the aerosol speciation in the whole atmospheric column, we compare the CHIMERE simulations outputs with CALIOP vertical cross-sections, (Winker et al., 2010). These cross-sections are constructed using to the CALIOP v4.10 product and the sub-type classification based on the studies of Omar et al. (2010) and Burton et al. (2015). This product is very useful since this is a realistic way to have an instantaneous evaluation of the aerosols layers, with their type and their altitude, as shown by (Chazette and Royer, 2017). Of course, the aerosol sub-type classification is built on




optical characteristics threshold and is not error free, as mentioned by Burton et al. (2013) and Huang et al. (2015). Limitations associated with this aerosol classification are described in Tesche et al. (2013).

| Code | CALIOP | CHIMERE |
|------|--------|---------|
| 0 | Not applicable | Not used |
| 1 | Clean marine | SALT |
| 2 | Dust | DUST |
| 3 | Pol. cont. or smoke | PM10ant - (EC+POM) |
| 4 | Clean cont. | PM10bio-SALT |
| 5 | Pol. dust | PPM |
| 6 | Elevated smoke | EC+POM |
| 7 | Dusty marine | DUST |

**Table 4.** *Correspondance between CALIOP 'optical indexes' and CHIMERE 'aerosol concentrations'.*

The equivalent of the CALIOP aerosol classification is obtained from CHIMERE using aerosol concentrations directly. The depolarization being not modelled, we have to find another way to reproduce the CALIOP classification. Some assumptions are made:

- The CALIOP terminology "elevated smoke" is difficult to evaluate in term of altitude. In Omar et al. (2010), it is stated that thin aerosol layers are stated as "clean continental" close to the surface or "smoke" if they are elevated. Over the ocean, all elevated non-dust aerosol layers are identified as smoke.

- CALIOP is particularly sensitive to clouds and Chen et al. (2012) noted that CALIOP often misidentifies aerosol as clouds. In Winker et al. (2013), "elevated layers" are considered as those up to 2 km above ground level.

- In this study, we make no difference between 'dust' and 'dusty marine': this is mineral dust.

- Many CALIOP profiles contains *"Not applicable"* values. It means that the detection algorithm was not able to affect an aerosol type. This is not the case with the model, where for each profile and each altitude, we are able to diagnose the major aerosol contribution. This means that the following figures for the model may appear to have more information.

The other hypotheses, to match as best as possible CALIOP 'optical indexes' with CHIMERE 'aerosol concentrations', are described in Table 4. The model species are, in general, directly linked to the CALIOP classification. As the model is able to separate PM from anthropogenic and biogenic origin, (Menut et al., 2013a), we use it to distinguish the "polluted continental" and "clean continental" aerosol layers. For the biomass burning emissions products, the 'smoke' is considered as the sum of Elemental Carbon (EC) and POM.

Comparisons between CALIOP and CHIMERE vertical transects are displayed in Figure 13 and for the 26 and 27th July 2014. We focus on these two days because (i) CALIOP data are available above the studied region, (ii) long-range transport of biomass burning is maximum at the end of the studied period.





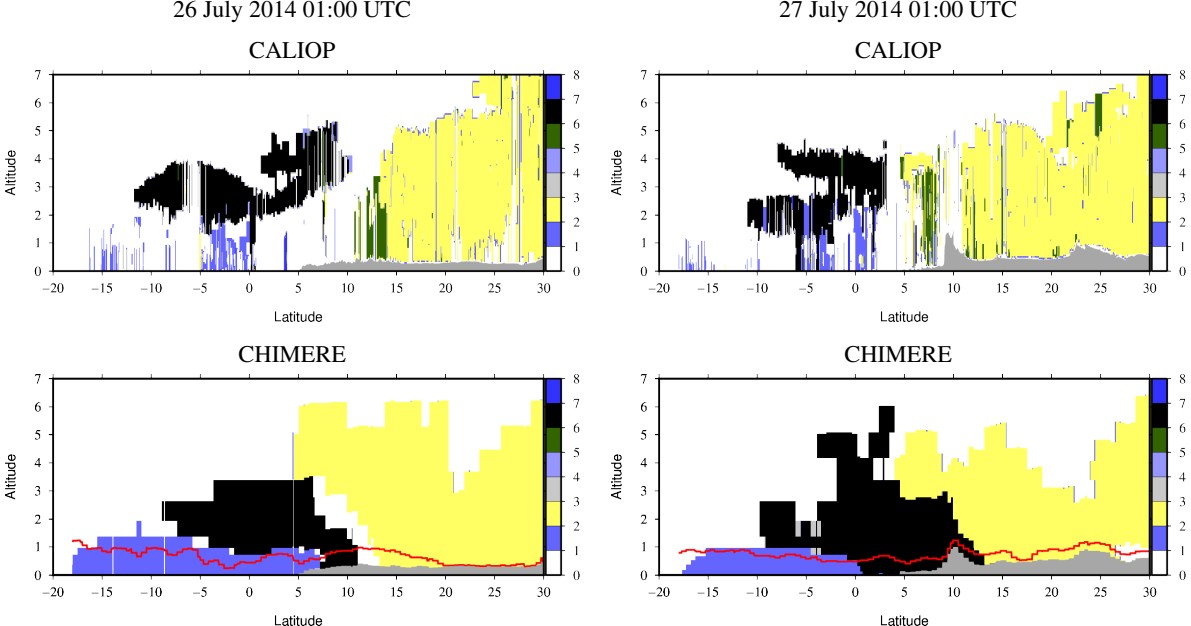

**Figure 13.** *Vertical cross-section of CALIOP aerosol types and comparison to the CHIMERE simulation. The colorbar is related to the CALIOP classification: (0:1) Not applicable, (1:2) Clean marine, (2:3) Dust, (3:4) Polluted Continental or smoke, (4:5) Clean continental, (5:6) Polluted dust, (6:7) Elevated smoke, (7:8) Dusty marine. For the model, the boundary layer height is superimposed in red.*

The two CALIOP ground-tracks are shown in Figure 1. The first result with this comparison is that the main air masses are well retrieved by the model: over land, the main aerosol is mineral dust and, over sea, sea-salt. Over sea and in altitude, the main aerosol type is due to biomass burning (noted as smoke). For the two days, the model is able to estimate the latitudinal extension of the smoke plume, from $\phi$=-15 to +10 $^o$N. For the vertical extension of smoke, the model under-estimates the top of the plume for the 26 July bit is correct for the 17 July. For this latter day, the double vertical plume is correctly reproduced by the model. The main difference between the model and the observations is that the smoke plume reaches the surface with the model but not in the observations. (Jethva et al., 2014) pointed out that in case of optically thick aerosol layer, the sensitivity of the CALIOP backscattered signal to the altitude of the base of the aerosol layer is strongly attenuated by the two-way transmission term. As a result, the operational algorithm may locate the base of the aerosol layer too high when it could actually be deeper and extend towards the surface. However, as noted previously, the CALIOP data are only for "elevated smoke", meaning that this is not because the CALIOP did not detect and attribute a smoke value above the marine layer that there is no smoke. In this sense, the model provides complementary insight about the plumes vertical extension.

Finally, this comparison with "instantaneous" measurements in the whole troposphere proves that the model is able to correctly estimate the location, latitude and altitude of the main studied aerosols. This improves our confidence in the model robustness.





## 6.6 Summary of results

In this section, the model results were compared to available measurements and for the whole period, from 1st May to 31 July 2014. Three model configurations, NoFIRE and FIREs with PR1 and PR2, were used to estimate the realism of the model as well as its uncertainty due to the vertical injection of biomass burning.

With the $PM_{10}$ surface measurements and the AOD time series, it was shown that the model is able to reproduce the mass and the optical properties of the aerosols. Located in the northern part of the modelled domain, these aerosols are mainly mineral dust. Using the IASI data, it was shown that the model is also able to reproduce the increase of CO in the total atmospheric column. The differences between the simulation NoFIRE and FIREs enable to have a first quantification of the additional CO due to biomass burning. Using the MODIS AOD products, it was shown that the largest values of AOD are mainly due to mineral dust for latitude up to $10^o$N and to biomass burning south of $10^o$N, conformed with the AERONET and modelled Angström exponent time series.

About the fires emissions uncertainty, it was shown that the results are very close between the two simulations PR1 and PR2. This is due to the fact that the fires are emitted over land and mainly during the convective period. In Central Africa, and for the studied months, the boundary layer height often reaches 3000 m or 4000 m AGL: the shape of the injection height profile plays a minor because the emissions are rapidly mixed in the boundary layer. After a few days, the emissions are then transported in the same way over long distances. For the next part of the paper, no significant differences having been found, the model results will be presented for the PR2 configuration only.

## 7 Impact on the coastal urbanized areas pollution

In this section, we focus on the atmospheric composition in coastal urbanized areas. The analysis is done with the model only, no data being available in the region and for the studied period. Results are presented for the sites Lagos (Nigeria) and Abidjan (Ivory Coast), representative of strongly urbanized coastal areas in the Guinean Gulf. The surface concentrations of tree chemical species are presented: (i) $O_3$, a secondary species both produced by anthropogenic, biogenic and fires emissions, (ii) CO, a gaseous species, primarily emitted by anthropogenic and fires emissions and $PM_{10}$, representative of the sum of aerosol produced by anthropogenic and natural sources.

### 7.1 Surface concentrations time series

Time series of surface concentrations of CO, $O_3$ and $PM_{10}$ are presented in Figure 14. The Figure presents the concentrations for NoFIRE and FIRE, as well as the difference (FIRE-NoFIRE). For the three species and the two locations, the impact of biomass burning appears after a few days. This impact has the same order of magnitude for the two sites, highlighting the long-range transport. The maximum contribution of the fire emissions is $\approx 150$ $\mu$g m$^{-3}$ for CO, $\approx 20$ $\mu$g m$^{-3}$ for $O_3$ and $\approx$ 5 $\mu$g m$^{-3}$ for $PM_{10}$. The contribution of fires appears as a smooth but steady increase (i.e. not pollution peaks): in case of





pollution alert, the number of exceedances will not be influenced by this biomass burning. Nevertheless, the increase would have a non negligible impact on human exposure.

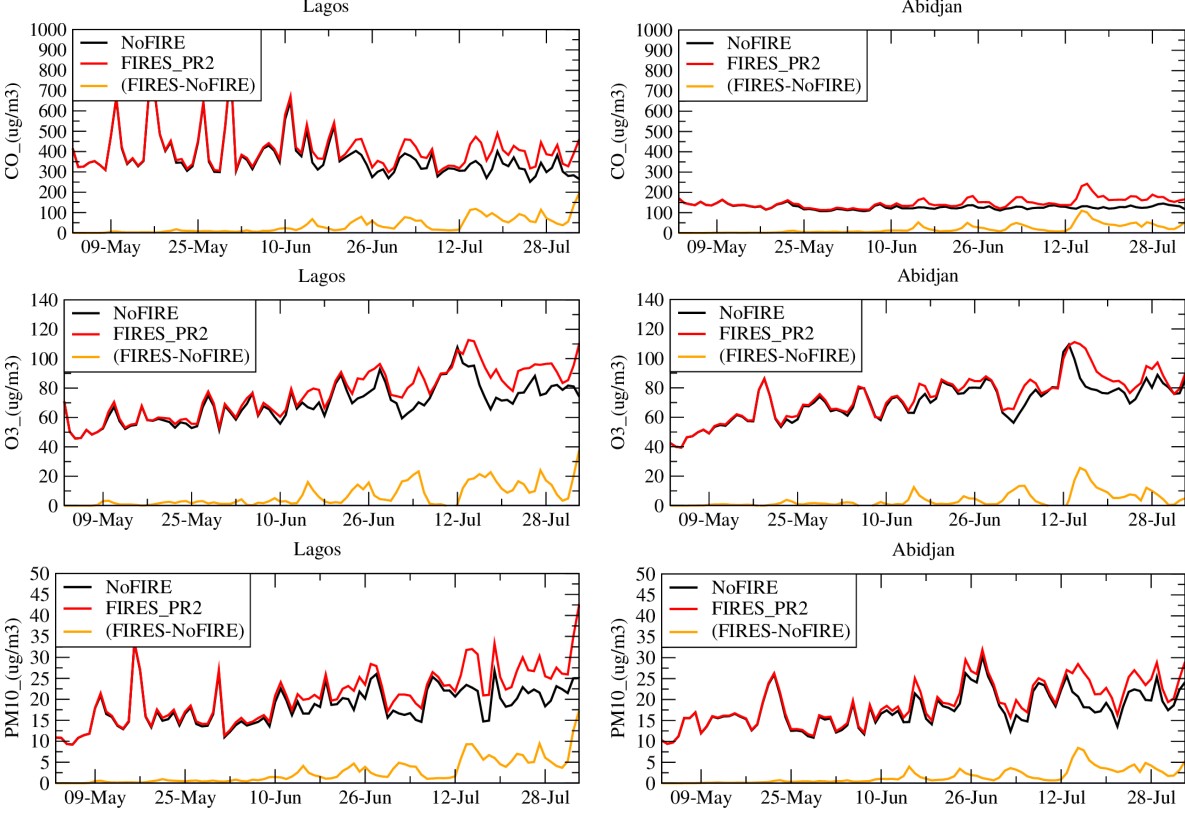

**Figure 14.** *Time series of surface concentrations (in $\mu g\,m^{-3}$) of CO, $O_3$ and $PM_{10}$. Results are presented for Lagos and Abidjan and for the simulations NoFIRE and FIRE PR2.*

## 7.2 Aerosol composition

10  The $PM_{10}$ is the cumulated mass of several aerosol types. With the model, it is possible to quantify the contribution of each aerosol. Results are presented in Figure 15 as differences between the simulations FIRE and NoFIRE in order to quantify the speciation of the additional amount due to biomass burning. Results are presented for Lagos and Abidjan.

The composition of the aerosol due to fires is mainly composed of POM and PPM. To a lesser extent, the aerosol is also composed of Ammonium, Sulfate and Secondary Organic Aerosol (SOA).





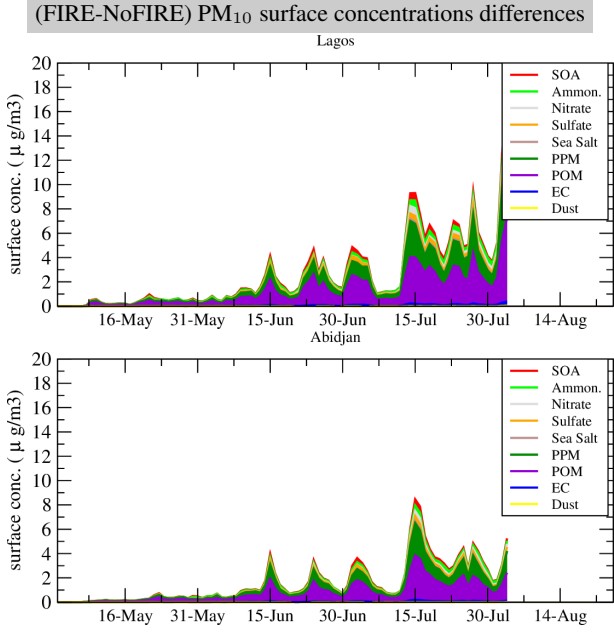

**Figure 15.** *Time series of PM$_{10}$ surface concentrations ($\mu g\, m^{-3}$) for the FIRE (PR2) simulation and for the difference (FIRE-NoFIRE). The speciation is presented for all aerosol species modeled with CHIMERE. The modeled data are daily averaged.*

## 8 Conclusions

This study examined the atmospheric composition during the summer 2014 (from May to July) in the region of the Guinean

Gulf. The main goal was to quantify the relative contribution of biomass burning emissions, occurring in Central Africa, on the aerosol, CO and O$_3$ surface concentrations in large urbanized areas such as Lagos and Abidjan. It was conducted in the framework of the DACCIWA European project, aiming to observe and model the dynamics, clouds and aerosol in the Guinean Gulf. Various natural and anthropogenic sources (local or remote) may induce large surface concentrations of gas and aerosol in the area.

The period was modeled with the meteorological model WRF and the chemistry-transport model CHIMERE. Several model configurations were used. First, and in order to know if the biomass burning pollutants may reach the Guinean Gulf cities (Lagos and Abidjan), a tracer experiment was performed. It was shown that, whatever the location of emissions in Central Africa, biomass burning always impacts the surface concentrations of the cities. Depending on the location of the emissions, the fire plumes may follow the coast to directly reach the cities, or are flowing to the east to be later caught in the Harmattan

circulation and then reoriented toward the cities. In order to reduce the uncertainty in the simulation due to the fires injection height, one of the unknown and uncertain parameter for this process, two simulations were performed with different vertical profiles for the emissions. Due to the strong convection in Central Africa, the emissions are completely mixed over land before



being transported over long distances: the modelled results thus showed no significant impact of this injection height on the modelled concentrations far from the sources.

The simulations with biomass burning emissions were analyzed by comparison to numerous datasets: CO with IASI, AOD with MODIS and AERONET, surface concentrations of $PM_{10}$ with the Sahelian Transect data and aerosol sub-type classification with CALIOP. It was shown that the model is able to reproduce the physical and chemical characteristics of the emitted gas and aerosol species due to biomass burning. In addition, and using the vertical information provided by CALIOP, it was shown that the localization and altitude of the several aerosol plumes (mineral dust and biomass burning) are correctly modelled.

Finally, and by comparison between a simulation without fires emissions (NoFIRE) and simulations with fire emissions (FIREs), a first quantification of the amount of additional pollutants in Lagos and Abidjan was presented. It was shown that biomass burning will induce a regular increase in pollutants surface concentrations during the whole studied period of the order of magnitude of $\approx 150 \ \mu g \, m^{-3}$ for CO, $\approx 20 \ \mu g \, m^{-3}$ for $O_3$ and $\approx 5 \ \mu g \, m^{-3}$ for $PM_{10}$. Using the modelled speciation, this additional amount was shown to be mainly composed of POM and PPM.

This study shows that the understanding of atmospheric pollution for urbanized areas in the Guinean Gulf region must take into account biomass burning in Central Africa. In this study, the model configuration was off-line and this may induce a bias in the result: the direct effect of dense biomass burning plumes may affect directly the convection in the region and the large amount of aerosols may also change the precipitations via by indirect aerosol effect. The next step will be to study this interaction by using an on-line coupled modeling system.

*Acknowledgements.* The research leading to these results has received funding from the European Union 7th Framework Programme
(FP7/2007-2013) under Grant Agreement no. 603502 (EU project DACCIWA: Dynamics-aerosol-chemistry-cloud interactions in West Africa). Thanks to the British Atmospheric Data Centre, which is part of the NERC National Centre for Atmospheric Science (NCAS), for the meteorological surface data used in this paper. S. Turquety acknowledges the French space agency (CNES) for financial support. The IASI CO data were provided by LATMOS/CNRS and ULB. The MODIS AOD datasets were acquired from the Level-1 and Atmosphere Archive and Distribution System (LAADS) Distributed Active Archive Center (DAAC), located in the Goddard Space Flight Center in
Greenbelt, Maryland (https://ladsweb.nascom.nasa.gov/). We thank Bernadette Chatenet, the technical PI of the Sahelian stations from 2006 to 2012, Béatrice Marticorena and Jean-Louis Rajot, the scientific co-PIs, and the African technicians who manage the stations. We thank the principal investigators and their staff for establishing and maintaining the AERONET sites used in this study. The CALIOP level 4.10 data, available at https://eosweb.larc.nasa.gov/, were obtained from the NASA Langley Research Center Atmospheric Science Data Center, which is greatly acknowledged.



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
