# Peer review of "Impact of biomass burning on pollutants surface concentrations in megacities of the Gulf of Guinea"

_Atmospheric Chemistry and Physics, 2017_

## Short Comment (SC1) · 4 Nov 2017

**Review of "Impact of biomass burning on pollutants surface concentrations in megacities of the Gulf of Guinea" by Menut et al. (2017)**

*Reviewed by Anne Swank*

*---This review was prepared as part of graduate program course work at Wageningen University, and has been produced under supervision of Prof Wouter Peters. The review has been posted because of its good quality, and likely usefulness to the authors and editor. This review was not solicited by the journal.---*

The study of Menut et al. quantifies the relative contribution of biomass burning emissions in Central and South Africa on the surface concentrations of CO, $O_3$ and $PM_{10}$ in urbanized areas in Southern West Africa. This is done with CHIMERE model simulations of biomass burnings and comparison with satellite and ground measurements data. Several (tracer) model simulations are performed and show that the biomass burning emissions do indeed have an impact on the surface concentrations in urban areas in Southern West Africa. This study contributes significantly to scientific research since it includes the air pollutants in the previously studied air masses transport in Africa, which is an important attribute since (anthropogenic) air pollutants are increasing and have an impact on human's health. The simulation of the model are well tough of and are compared properly to available data. Hence this study fits to the reader's interest of Atmospheric Chemistry and Physics. However, the answer on the main research question about the relative contribution of biomass burning emissions cannot be found in this paper, which can be attributed to the absence of background information and clear explanation between the connection of observed and simulated air pollution values. These quantified relations are sometimes missing and some figures could be shown in another way. Therefore I recommend some major changes in this paper, at least to answer the main research question and on some other aspects, prior to publishing this paper in ACP.

*Major arguments:*

The paper does not describe clearly how the biomass burning emissions are contributing to the total air pollution in the simulations and observed concentrations, where the research question is how the emissions contribute relatively. The results of the CHIMERE model simulations as illustrated in Figure 14, clearly with the simulation differences, show the maximum increase in surface concentrations of CO, $O_3$ and $PM_{10}$ of about 150 µg m$^{-3}$, 10 to 20 µg m$^{-3}$ and 5 µg m$^{-3}$ respectively. Concluded from these values is that the exceedance of pollution alerts will not by influenced hereby, but the impact on human exposure is not negligible. In the study of Adon et al. (2016) the concentration of $O_3$ in West African urban environments ranges from 5.5 to 7.7 ppb (equal to µg/m$^3$), which indicates that an increase of 5 µg/m$^3$ would have an influence. From the study of Antonel & Chowdhury (2014) the $PM_{10}$ concentration are in range from 60 to 140 µg/m$^3$. I believe that the increase in $PM_{10}$ due to biomass burning will therefore not have a large influence on the pollution alert. But, from my view it is necessary to include these reference (background) values of air pollutants in the urban areas in SWA, if necessary include observations form previous WAM years, and concentrations when the air pollution alert will be exceeded. The same accounts for the air pollution concentrations which have an impact on human exposure, when certain air pollutants do have a large impact more attention could be paid to these in the study, serving the purpose of the DACCIWA project. With the reference values included the relative contribution of biomass burning emissions on air pollution and its impact can be determined.

In the explanation for the TRC experiment in paragraph 3.5 is stated that the tracer emission flux is injected constantly without including the diurnal cycle, which produces the results as shown in Figure 7 and 8. However, in the study of Parker et al. (2005) is stated that the diurnal cycle has implications on the mixing of trace gases and aerosols between the surface layer and free troposphere. The vertical mixing occurs to be most efficient at night, which indicates that for the tracer experiment the diurnal cycle should be taken into account. This is confirmed by the study of Gilge et al. (2010), which indicates

that an increase in the vertical mixing within the free troposphere could influence the air pollution levels in the lower free troposphere, with implications to the boundary layer. As consequence different vertical mixing profiles due to the diurnal cycle will probably have an influence on the tracer emission fluxes and transport. However, in section 3.4 of this study is indicated that the two different vertical mixing profiles provide the same results for the transport of the tracer and that only the profile of PR2 will be discussed. Wouldn't the transport be affected by different vertical mixing profiles when the tracer emission experiences a diurnal cycle? It might be good to discuss this in the article with results of the tracer experiment including a diurnal cycle for the two different vertical profiles.

The simulations in this study to quantify the biomass burning emissions are performed with the CHIMERE model. In the article is stated that comparison between satellite data, ground observations and simulations of the CHIMERE model show that the output model results are robust. This is illustrated for instance as time series in Figure 11 for the CO concentrations. What I miss in this article is why the air pollution concentrations are simulated with the use of the CHIMERE instead of another air chemistry model. In the study which Solazzo et al. performed in 2017, but with the CHIMERE model version of 2013, is stated that the CHIMERE model as its performance is studied in Europe, not always simulates the air pollutant concentration correctly. When the ozone lateral boundaries are changed, a shift is visible in the ozone diurnal cycle of the CHIMERE model with significant impact, which could indicate a flaw in the PBL dynamics. A positive bias in the ozone concentration simulated by CHIMERE is also concluded from the study of van Loon et al. (2017). Besides, the error of $NO_2$ impacts influence the ozone error significantly. These problems do not seem to be solved as I read from the article of Mailler et al. (2017) which describes the 2017 updates of CHIMERE. Due to the flaws in the model on ozone specifically I am not convinced that CHIMERE is the model to use in this study at this moment. An explanation about the simulated ozone concentrations or the relation between the ozone simulations and observations would help me to understand the choice to use CHIMERE. Besides, including some (necessary) correlations between observations and CHIMERE simulations for at least $O_3$, aerosol subtype and CO would convince the reader of the robustness of the model, similar as Table 2 and 3 for AOD and $PM_{10}$, next to only the quantitative differences in Figure 11 and 13 for instance.

*Minor arguments:*
The conclusion from this paper is that the increase in air pollutants is mainly related to PMM and POM, indicating biomass burning. This is stated in part 7.2 of results, however I cannot find (neither in literature) whether this statement can be related to a real life scenario. Quantitative argumentation and references should be included on this matter.

In chapter 3 is mentioned that CHIMERE reads (WRF and) the surface emissions to simulate the chemical concentration, it is not described how these are obtained or can I assume that these are obtained by MODIS data? A brief explanation would be adequate.

In the article the output of the simulation of the CHIMERE model in the urban areas are quantified. However, it is not stated clearly how the values are obtained. Are these point measurements or averages over the whole urban area?

It is a bit unclear to me, whether all fires for the PR2 simulations are estimated on the same Hp or are estimated per individual fire (as described in paragraph 3.4 page 9)? I believe that the magnitude of fires differs and results in different Hp which are influencing the transport of air particles. The explanation could be improved.

*Minor issues:*

*p1, abstract:* in the abstract the main goal of the study is framed differently than throughout the paper

*p3, Table 1:* the caption of Table 1 and elsewhere throughout the manuscript is situated below the table, where it should be placed above

*p4, Figure 1:* adding a legend to the figure would help the reader to understand it in one glimpse

*p9, line 8:* probably you mean the 'the transport of fire emissions' instead of 'transport of fires"

*p9, line 27:* "than" should be replaced by 'as'

*p10, line 11:* 'complementary information is provided' should be stated here

*p10:* it would be logical to place Figure 1 below chapter 3 since it elaborates on the different areas

*p16, Figure 9:* a clearly visible title should be provided for each output figure including the unit of the values and the studied variable. Besides, wouldn't it be clearer to the reader if the simulation deviation between simulated and observed values is visualized?

*p19, last line:* "And" can be left out

*p25, line 14:* after "a minor" 'role' should be stated

*p25, chapter 7:* in the first paragraph it is forgotten to include "(iii)"

*p27, line 6:* in the introduction of the conclusion $PM_{10}$ is not mentioned

*p28, line 18:* choose "by" or "via" and include 'the'

*Literature list:*

Adon, M., Yoboué, V., Galy-Lacaux, C., Liousse, C., Diop, B., Gardrat, E., ... & Jarnot, C. (2016). Measurements of NO 2, SO 2, NH 3, HNO 3 and O 3 in West African urban environments. *Atmospheric Environment*, *135*, 31-40.

Antonel, J., & Chowdhury, Z. (2014). Measuring ambient particulate matter in three cities in Cameroon, Africa. *Atmospheric environment*, *95*, 344-354.

Gilge, S., Plass-Dülmer, C., Fricke, W., Kaiser, A., Ries, L., Buchmann, B., & Steinbacher, M. (2010). Ozone, carbon monoxide and nitrogen oxides time series at four alpine GAW mountain stations in central Europe. *Atmospheric Chemistry and Physics*, *10*(24), 12295-12316.

Mailler, S., Menut, L., Khvorostyanov, D., Valari, M., Couvidat, F., Siour, G., ... & Colette, A. (2017). CHIMERE-2017: from urban to hemispheric chemistry-transport modeling. *Geoscientific Model Development*, *10*(6), 2397.

Parker, D. J., Burton, R. R., Diongue-Niang, A., Ellis, R. J., Felton, M., Taylor, C. M., ... & Tompkins, A. M. (2005). The diurnal cycle of the West African monsoon circulation. *Quarterly Journal of the Royal Meteorological Society*, *131*(611), 2839-2860.

Solazzo, E., Hogrefe, C., Colette, A., Garcia-Vivanco, M., & Galmarini, S. (2017). Advanced error diagnostics of the CMAQ and Chimere modelling systems within the AQMEII3 model evaluation framework. *Atmos. Chem. Phys.,17, 10435-10465*.

Van Loon, M., Vautard, R., Schaap, M., Bergström, R., Bessagnet, B., Brandt, J., ... & Jonson, J. E. (2007). Evaluation of long-term ozone simulations from seven regional air quality models and their ensemble. *Atmospheric Environment*, *41*(10), 2083-2097.

---

## Short Comment (SC2) · 6 Nov 2017

"This review was prepared as part of graduate program course work at Wageningen University, and has been produced under supervision of Prof Wouter Peters. The review has been posted because of its good quality, and likely usefulness to the authors and editor. This review was not solicited by the journal."

Please also note the supplement to this comment:
https://www.atmos-chem-phys-discuss.net/acp-2017-852/acp-2017-852-SC2-supplement.pdf

[Figure]

**Supplement:**

**ITEE part III Peer Review**
By Jolanda Theeuwen

**'Impact of biomass burning on pollutants surface concentrations in megacities of the Gulf of Guinea' by Menut et al.**

This research is done to quantify the relative contribution of biomass burning in southern and central Africa to the atmospheric composition in mega-cities in the Gulf of Guinea. First the WRF model is used to simulate the meteorological input data for the CHIMERE model in the period of May-July 2014. Second, tracer experiments, releasing tracer at two different heights and two different locations, are done using the CHIMERE model. Third the long-range transport of the pollutants is analysed with three situations: no fires, fire emissions injected in the lower atmosphere and fire emissions injected in the upper atmosphere. From this research can be concluded that first, the biomass burning in this area is contributing to the atmospheric composition in cities in the Gulf of Guinea. Second, transport of the pollutants is different for different emissions heights, however after a couple of weeks the concentrations are similar for the different injections. Third, the biomass burning resulted in a maximum increase of 150, 20 and 4 $\mu gm^{-3}$ for respectively CO, $O_3$ and $PM_{10}$ (Particulate Primary matter and Particular Organic matter).

This research is interesting as it is a contribution to the DACCIWA project. The results will add new knowledge on what the influence of the biomass burning is on atmospheric pollution in the area of the Guinea Gulf. The DACCIWA project is set up to analyse the impact of the atmospheric composition on populations health. The project and therefore a contribution to this research, are of social importance because of the highly polluted megacities in western Africa.

This research is a general implication of atmospheric modelling and therefore, I believe it fits the scope of this journal well. Both the physics, the meteorological conditions, and the chemistry, the transport of chemicals, are simulated in this research. These themes fit the subject of the journal well.

The poor use of grammar negatively influences the clarity of writing in this article for me. The article is very long which also has a negative influence on the clarity. However, with the help of summaries in between different sections the article is made clearer for me.

I think the work done in this research is not completely correct. If I would be doing this research I would reconsider the simulated period. Additionally, I would add an uncertainty indication in the results to account for observation and simulation errors. However, I think the overall steps in this research are chosen well and support the conclusion. Because is shows some flaws I recommend the editor not to accept this article yet. However, after some changes I definitely think this article will be a valuable addition to the journal. Some examples of changes are reformulating the introduction; the need and scope need to be introduced better, and adding an uncertainty indication in the conclusion. These examples are explained more thoroughly further on in this peer review.

**Strengths**
As mentioned before I believe the overall steps in this research are chosen well and support the conclusions. The order of the different parts in this research is correct and each step gives value to the research.

Furthermore, I think that the authors put a great effort in creating structure in this article. In the introduction, the content of the article is clearly explained also the approach of the research is explained shortly. Also the summaries after each section contribute to the overall structure of the article. These summaries allow the reader to put everything into place. The conclusions drawn here are eventually put together in the final conclusion. Throughout the article bullet points are used for summations which also makes the article more readable.

**Major points of improvement**

The first point of improvement in this research is the choice of the modelling period. The authors reference to Barbosa et al. 1999 (p6 line 28) to explain their choice of the modelling period. The authors state that the fires in central Africa start in April and peak in July and therefore they modelled the months May-July. In the referenced article, I found that fires in Africa on the southern hemisphere indeed start in April and end around October. However, on the northern hemisphere the fire burning activity begins in November. The, in this research investigated, area covers both the northern and the southern hemisphere and therefore I believe that the performed research is incomplete. No specific time period is mentioned in the scope of this research and therefore the goal is to analyse the 'general' contribution of biomass burning emissions. The authors give insight in the atmospheric composition for a period of three months and therefore it does not completely fit the scope of this research I believe. For this research to fit the scope I think that the authors should give insight in fire burning throughout an entire year by modelling also the peak events on the northern hemisphere.

It could be possible that due to meteorological conditions the fires in the northern hemisphere do not influence the pollutant concentrations in the megacities in the studied area. However, if this would be the case the article would be stronger if sufficient evidence would be given. I recommend the authors to look into the atmospheric dynamics in the northern hemisphere by setting up a WRF simulation for this area and use the gained output for additional tracer experiments. This should be done for the months November-January. If the emissions in the northern hemisphere haven an important role in the atmospheric composition of the studied megacities the entire research should be repeated for the months November-January to fit the scope.

Second, in the article is mentioned that the simulated meteorological conditions lead to differences between model and observation, comparing AOD CHIMERE and Modis (p 17 line 10). It is however not discussed what the approximate contribution of this error is to the simulated result. Throughout the entire article this is the only reference to the possible influence on the model simulations by the deviating meteorological input conditions. I think the article will improve if uncertainties are included throughout the article.

Meij et al (2009) shows that different meteorological conditions can have a different influence on the modelled $O_3$ and $PM_{10}$ concentrations. In this research two simulations are used. One simulation is overestimating the precipitation and the other one is underestimating the precipitation. In Meij et al (2009) the conclusion is drawn that for $PM_{10}$ in January the concentration differs with a factor of 1.6 for the different simulations. This shows that meteorological conditions indeed have a significant influence on the concentrations. According to the reviewed article the change in $PM_{10}$ by biomass burning is $5\mu gm^{-3}$. I believe an increase or decrease of a factor 1.6 is a significant difference and should therefore be investigated. In the reviewed article the authors state that the meteorological conditions are deviating a bit due to the difficulty of simulating these highly variable processes. However, in the results it is mentioned that some results are quite similar and they are even called 'realistic'. The authors show that both overestimations and underestimations are made in the model. Meij et al (2009) shows that this influences the CHIMERE results.

To improve on this point the author should include the uncertainty more often and more in detail. The author has two options to implement the uncertainty. First, the author can include it by discussing the possible influences of the deviating meteorological conditions on further simulations in more detail. The authors should then discuss for every result the uncertainty of the simulations but also the uncertainty in the observations. These discussions then need to be combined at the final conclusion to give an educated guess of the uncertainty in the research. The second option is to redo the experiments with different settings. An ensemble run can be done to quantify the uncertainty. When the authors perform an ensemble run it will allow them to include error bars in their conclusion on the increase of $O_3$, CO and $PM_{10}$ concentration. Including these error bars will show that the authors looked

critically at their own results which can be seen as valuable. It will also make the article more credible. Because the conclusion gives a clear quantification of the contribution of biomass burning I personally believe option two fits this research well. However, discussing the insecurities of the performed research is always a good addition to scientific research. This discussion clearly indicates where improvements are needed and where further research is necessary.

The third point of improvement is the content of the introduction and how it fits to the conclusion. In the introduction is stated that this study is done for the DACCIWA project to prepare for a field campaign in June/July 2016. The paper also states that the aim of the DACCIWA project is to define the variability of the atmospheric composition and its impact on West African climate and health. It is however not stated how exactly the results will contribute to this project and the field campaign. I think the content of the current introduction does not clearly describe the need for this research. Also is the aim of the DACCIWA project clearly described but is the aim of this research not strongly present. Without this information, it makes it more difficult to put the results into context. However, this information is important for the reader. This type of information motivates me to keep reading an article. The impact of the paper is also smaller without a clear explanation of the need for this research.

Additionally, the introduction proposes the research question which does not completely fit the conclusions. This research is set up to quantify the relative impact of biomass burning in central Africa on the atmospheric concentrations of $O_3$, CO and $PM_{10}$ in the Guinea Gulf area. The conclusion does quantify the absolute impact of biomass burning on these concentrations but not the relative contribution. This information could be relevant defining the problem and dealing with the problem. To improve populations health and climate the project should focus on emissions that have a large influence on the pollutant concentration. To know which factors have an important contribution to the atmospheric composition, the total concentrations of $O_3$, CO and $PM_{10}$ need to be known. To give an indication to the author on how this additional information can be a valuable for the article I looked up some numbers. In Lagos, one of the studied cities, the $PM_{10}$ concentration during the morning is 476.35 $\mu gm^{-3}$. During the afternoon, this concertation is 454.60 $\mu gm^{-3}$ (Odekanle et al. 2016). The increase due to biomass burning is approximately 5 $\mu gm^{-3}$ this is less than 1%. Concentration of $O_3$ is approximately 50 ppb and the concentration of CO is between 75 and 300 ppb (Jambert et al. 2017). These concentrations are equal to 100 $\mu gm^{-3}$ $O_3$ and 80.88-343.5 $\mu gm^{-3}$ CO. Respectively the influence of biomass burning on the concentrations is 10-20% and 44-185%. These parts are much larger than the contribution to the $PM_{10}$ concentration. With these percentages, it is clearer what the relative impact of biomass burning is in polluting the atmosphere.

To improve the introduction the author should first include a part that describes the exact contribution of this research to science and also the aim. Additionally, the author should include a part on the average $O_3$, CO and $PM_{10}$ concentrations in the studied area. This information can then be used in the conclusion to add a part that describes what the relative contribution of biomass burning is to the concentration of these three chemical components. To make the introduction and conclusion fit both need some adjustments.

**Minor points of improvement**
*Minor points concerning figures and tables*
I believe figure 2 is not a valuable addition and can therefore be taken out completely.

Data is missing in figures 2 and 13. In section 2 is written: 'Finally, note that, for chemistry, there is a lack of in-situ surface measurements for this region and during the studied period.' This could direct to the chemistry in figure 2. However, I do not understand if the authors indicate the missing data in

figure 13 with this as well. Maybe the authors could specify which observation techniques have missing data instead of saying there is data missing in chemistry.

For the description of figure 5 please include that the observations are presented with dots. The text should include a reason why the observations in this figure are different for 15 May and 15 July.

For figures 6, 7, 10, 12 and 14 there is some overlap between the legend and the graphs.

Please present figure 7 (Lope) and 12 in a cleaner way. The figures now are difficult to read due to different scaling and overlap.

In figure 13 I would put the legend in the image instead of in the description of the figure. I think this will make the figure easier to read and it is more consistent compared to other figures. Additionally, a lot of data is missing in this figure please discuss this.

Table 2 was a bit difficult to read for me. The description is incomplete, it says the table shows the correlation between the observations and the model for de AOD. The description should also include that the observed AOD and the modelled AOD are included in this table.

For table 3 please also include in the description the observed and modelled values for $PM_{10}$ surface concentrations are included in the table. Please also include unities in the first line of the table. Finally, the description says that the RMSE is given in the table but only the bias is presented.

*General minor points*
The title is very general and therefor does not state the important conclusion. A possibility is: 'The significant impact of biomass burning on pollutants surface concentrations in megacities of the Gulf of Guinea.' If this title is used it supports the conclusion that biomass burning in central Africa must be taken into account when understanding the atmospheric pollution in the Guinea Gulf area.

In the introduction please include a reference on previous research that is similar to this one. Only for every separate part references of previous research are included. I think it will add value when one (or more) reference(s) on modelling fire emissions making use of both WRF and CHIMERE is (are) added. I think it will make the research more reliable because in this article is stated that wild-fire fluxes are in chemistry transport models, one of the most uncertain sources (p6 line 29-30).

The last two paragraphs in the introduction give a clear view on the structure of the research and article. Please include which models are used for each step. I think this will make it easier for the reader to understand already the first time reading the article, which model was used to investigate each step.

IASI CO columns are recovered using the FORLI algorithm. It is unclear to me whether the obtained data is filtered and if so, how is it filtered? Pommier et al. (2017) shows that not all measurements are reliable and the unreliable measurements are filtered out using the RMSE and bias. (p19 line 13-14)

Please add how the vertical levels are ordered in WRF (p 4 line 13).

*Minor points in use of grammar*
Throughout the entire paper the level of grammar is poor and it disturbs the readability. Some examples are presented below.

- Typos: p 2 line 27-28: groud-based stations --> ground-based stations
- Syntax: p3 line 3: the model validation, the discussion on the biomass burning.. --> the model validation and/or the discussion on the biomass...

- Misuse of word: P 17 line 10 this is possible --> and therefore it is possible
  P23 line 20: Figure 13 and for the.. --> Figure 13 for the...

- Plural: p17 line 21: fires --> fire
  P23 line 18: emissions products --> emission products

- Conjugations: p23 line 12: contains --> contain
- Sloppyness: p 25 line 26: PM10 should be indicated with a count --> and iii) PM10

**References**

Barbosa, Paulo Marinho, et al. "An assessment of vegetation fire in Africa (1981–1991): Burned areas, burned biomass, and atmospheric emissions." Global Biogeochemical Cycles 13.4 (1999): 933-950.

Jambert, Corinne, et al. "Observations of biogenic isoprene emissions and atmospheric chemistry components at the Savé super site in Benin, West Africa, during the DACCIWA field campaign." EGU General Assembly Conference Abstracts. Vol. 19. 2017.

Meij, A. de, et al. "The impact of MM5 and WRF meteorology over complex terrain on CHIMERE model calculations." Atmospheric Chemistry and Physics 9.17 (2009): 6611-6632.

Odekanle, E. L., et al. "Personal exposures to particulate matter in various modes of transport in Lagos city, Nigeria." Cogent Environmental Science 2.1 (2016): 1260857.

Pommier, Matthieu, Cathy Clerbaux, and Pierre-Francois Coheur. "Determination of enhancement ratios of HCOOH relative to CO in biomass burning plumes by the Infrared Atmospheric Sounding Interferometer (IASI)." Atmospheric Chemistry and Physics 17.18 (2017): 11089-11105.

---

## Author Comment (AC1) · 6 Nov 2017

Dear Anne Swank,

Thanks a lot for your comments on our work. I have to say it was a surprise to have this comment! When ACP was created, the novelty (compared to other peer-reviewed journals) was to have comments apart from 'official' reviews. But if you are looking at the numerous papers currently submitted, you can see that the counter is often 0. So, this is a good thing to have this additional comment and we are happy and proud that our work, only submitted (and thus open to improvements), may be used for courses. Of course, we consider this is just a comment and I don't consider the fact that you have a judgment of what has to be published or not. You will find here some answers to your most important questions.

*Anne Swank text (in italic):*
*"The study of Menut et al. quantifies the relative contribution of biomass burning emissions in Central and South Africa on the surface concentrations of CO, O3 and PM10 in urbanized areas in Southern West Africa. This is done with CHIMERE model simulations of biomass burnings and comparison with satellite and ground measurements data. Several (tracer) model simulations are performed and show that the biomass burning emissions do indeed have an impact on the surface concentrations in urban areas in Southern West Africa. This study contributes significantly to scientific research since it includes the air pollutants in the previously studied air masses transport in Africa, which is an important attribute since (anthropogenic) air pollutants are increasing and have an impact on human's health. The simulation of the model are well tough of and are compared properly to available data. Hence this study fits to the reader's interest of Atmospheric Chemistry and Physics. However, the answer on the main research question about the relative contribution of biomass burning emissions cannot be found in this paper, which can be attributed to the absence of background information and clear explanation between the connection of observed and simulated air pollution values. These quantified relations are sometimes missing and some figures could be shown in another way. Therefore I recommend some major changes in this paper, at least to answer the main research question and on some other aspects, prior to publishing this paper in ACP."*

As you write, the main topic of the paper is to quantify the "relative" contribution of biomass burning and this is done in the paper. During this period and over the studied region, there was no available measurements data (except those used in the paper): compared to studies done in Europe, China or South-Americ, there is no air quality network and no surface measurements of pollutants such as ozone, nitrogen oxides. Thus, we consider that the model is able to simulate the background concentrations, knowing that it was used over several other regions, with the same amount of information (HTAP anthropogenic emissions, Apiflame biomass burning emissions, Megan model biogenic emissions etc.). See for example the recent publications of [Marecal et al., 2015], [Real et al., 2015], [Menut et al., 2015], [Mallet et al., 2016], [Bessagnet et al., 2016], [Menut et al., 2016], [Vivanco et al., 2017]. These references are reported at the end of this answer. But, this is right, we have to make the hypothesis that, in absence on local surface measurements, the background concentrations have the correct order of magnitude and, thus, we can consider that the differences between the two simulations (without and with biomass burning emissions) also have a correct order of magnitude.

*Anne Swank text (in italic):*
*Major arguments: The paper does not describe clearly how the biomass burning emissions are contributing to the total air pollution in the simulations and observed concentrations, where the research question is how the emissions contribute relatively. The results of the CHIMERE model simulations as illustrated in Figure 14, clearly with the simulation differences, show the maximum increase in surface concentrations of CO, O3 and PM10 of about 150 µg m-3, 10 to 20 µg m-3 and 5 µg m-3 respectively. Concluded from these values is that the exceedance of pollution alerts will not by influenced hereby, but the impact on human exposure is not negligible. In the study of Adon*

*et al. (2016) the concentration of O3 in West African urban environments ranges from 5.5 to 7.7 ppb (equal to μg/m3), which indicates that an increase of 5 μg/m3 would have an influence. From the study of Antonel & Chowdhury (2014) the PM10 concentration are in range from 60 to 140 μg/m3. I believe that the increase in PM10 due to biomass burning will therefore not have a large influence on the pollution alert. But, from my view it is necessary to include these reference (background) values of air pollutants in the urban areas in SWA, if necessary include observations form previous WAM years, and concentrations when the air pollution alert will be exceeded. The same accounts for the air pollution concentrations which have an impact on human exposure, when certain air pollutants do have a large impact more attention could be paid to these in the study, serving the purpose of the DACCIWA project. With the reference values included the relative contribution of biomass burning emissions on air pollution and its impact can be determined.*

As you say, the quantification is presented in Figure 14 and in the text. The main question of the article has thus its answer. And there is not in our text the "total air pollution" but "pollutants". This is not the same thing. We are giving an asnwer for ozone, CO and PM10, three of the main pollutants, but, of course, not all pollutants. The model takes into account all possible emissions (and this is not the case of many regional models currently used in the community). The differences presented at the end of the paper are thus really representing the additional amount due to fires, in addition to all other sources.

About the background values, even if we are studying the same SWA region, this is not reasonable to compare different areas, different periods. The meteorology and the biomass emissions are very changing with time, from day to day and the pollutants plumes may be very different from a country to another one. So, this is not correct to directly use previous studies done for other period or locations and use the values to directly conclude something. About exceedances, there is periods where surface concentrations exceed threshold, but we can not extrapolate past values on our period.

*Anne Swank text (in italic):*
*"In the explanation for the TRC experiment in paragraph 3.5 is stated that the tracer emission flux is injected constantly without including the diurnal cycle, which produces the results as shown in Figure 7 and 8. However, in the study of Parker et al. (2005) is stated that the diurnal cycle has implications on the mixing of trace gases and aerosols between the surface layer and free troposphere. The vertical mixing occurs to be most efficient at night, which indicates that for the tracer experiment the diurnal cycle should be taken into account. This is confirmed by the study of Gilge et al. (2010), which indicates that an increase in the vertical mixing within the free troposphere could influence the air pollution levels in the lower free troposphere, with implications to the boundary layer. As consequence different vertical mixing profiles due to the diurnal cycle will probably have an influence on the tracer emission fluxes and transport. However, in section 3.4 of this study is indicated that the two different vertical mixing profiles provide the same results for the transport of the tracer and that only the profile of PR2 will be discussed. Wouldn't the transport be affected by different vertical mixing profiles when the tracer emission experiences a diurnal cycle? It might be good to discuss this in the article with results of the tracer experiment including a diurnal cycle for the two different vertical profiles."*

You probably missed this part in the paper: "The fluxes being daily estimated, a diurnal profile is applied where 30% of the daily is redistributed during the night (18:00 to 8:00 LT-local time) and 70% during the day, close to values usually chosen in biomass burning model studies, (Zhang et al., 2012)."

Biomass burning emissions are estimated using satellite data. But this is not a geostationary satellite and there are no measurements every hour. We thus have to make hypothesis about the diurnal cycle. Even if our hypothesis is not perfect, this corresponds to the state of the art for these emissions.

*Anne Swank text (in italic):*
*"The simulations in this study to quantify the biomass burning emissions are performed with the CHIMERE model. In the article is stated that comparison between satellite data, ground observations and simulations of the CHIMERE model show that the output model results are robust. This is illustrated for instance as time series in Figure 11 for the CO concentrations. What I miss in this article is why the air pollution concentrations are simulated with the use of the CHIMERE instead of another air chemistry model. In the study which Solazzo et al. performed in 2017, but with the CHIMERE model version of 2013, is stated that the CHIMERE model as its performance is studied in Europe, not always simulates the air pollutant concentration correctly. When the ozone lateral boundaries are changed, a shift is visible in the ozone diurnal cycle of the CHIMERE model with significant impact, which could indicate a flaw in the PBL dynamics. A positive bias in the ozone concentration simulated by CHIMERE is also concluded from the study of van Loon et al. (2017). Besides, the error of NO2 impacts influence the ozone error significantly. These problems do not seem to be solved as I read from the article of Mailler et al. (2017) which describes the 2017 updates of CHIMERE. Due to the flaws in the model on ozone specifically I am not convinced that CHIMERE is the model to use in this study at this moment. An explanation about the simulated ozone concentrations or the relation between the ozone simulations and observations would help me to understand the choice to use CHIMERE. Besides, including some (necessary) correlations between observations and CHIMERE simulations for at least O3, aerosol subtype and CO would convince the reader of the robustness of the model, similar as Table 2 and 3 for AOD and PM10, next to only the quantitative differences in Figure 11 and 13 for instance."*

There are several reasons to use CHIMERE. Some are subjective, the others objective. CHIMERE is a research model and, as all models, it is not perfect. The first subjective point is we are the developers of the model, therefore we are using it. The studies you are citing (Solazzo, Van Loon) were done by users of the model (there is about 300 persons using the model we are developing at the lab). A part of our work is to improve the model constantly and offer to the scientific community a new version each year. The second subjective point is that, being the developers, we can test parameterizations, schemes and modifying the code as we want. The objective points is that we are confident in our results. This is proved by many other studies you are not citing: see the results in the references cited at the end of this letter. You can also see the CHIMERE web page with all publications and you will see that for a large part of studies, CHIMERE is robust for the ozone concentrations modeling. And this is for this reason that the model is used for forecast in many Air Quality Networks in Europe, for the French official Air Quality Forecast called PREVAIR and is one of the 8 models selected to run daily the air quality forecast in the famework of CAMS Copernicus.

Another important point: there is no "good" or "not good" model. A chemistry-transport model is the sum of many processes (meteorology, emissions mixing, deposition) and all existing regional models have strengths and weaknesses… But knowing the status of the other available models, we are not sure to have better results.

*Anne Swank text (in italic):*
*"Minor arguments: The conclusion from this paper is that the increase in air pollutants is mainly related to PMM and POM, indicating biomass burning. This is stated in part 7.2 of results, however I cannot find (neither in literature) whether this statement can be related to a real life scenario. Quantitative argumentation and references should be included on this matter."*

You can read the paper explaining how we calculate the biomass burning emissions : the Apiflame model. It was also developed in our research team and you will see the emitted model species, PPM and POM, are some of these species. The reference to Apiflame is already in the paper.

*Anne Swank text (in italic):*
*« In chapter 3 is mentioned that CHIMERE reads (WRF and) the surface emissions to simulate the chemical concentration, it is not described how these are obtained or can I assume that these are obtained by MODIS data? A brief explanation would be adequate. »*

No, the emissions can not be obtained from satellite measurements. An emission flux and an atmospheric concentration (even if it is close to the surface) are completely different : between the two, you have the mixing, the chemistry, the deposition. All explanations about the emissions are provided in the section 3.2, with many details in (Menut et al., 2013) about all these processes.
For the anthropogenic emissions, explanations and references are at the end of section 3.2, for mineral dust emissions, this is the section 3.3 and for the biomass burning emissions, the section 3.4.

*Anne Swank text (in italic):*
*"In the article the output of the simulation of the CHIMERE model in the urban areas are quantified. However, it is not stated clearly how the values are obtained. Are these point measurements or averages over the whole urban area?"*

Yes, you are right, we can add some lines to better explain how we select the location to compare model and observations. This is a good point and we will revise the manuscript with this suggestion. The principle is as follows : when you have a surface stations (such as MIDAS or AERONET), you have the exact location in longitude and latitude. The model has an horizontal resolution of 60km in this study. We use the four model points around the location and we calculate the corresponding concentrations using a bilinear interpolation. This is the same for the cities : but, in this case, having no specific location, we are using the center of the city to know what grid cell to use for our interpolation.

*Anne Swank text (in italic):*
*"It is a bit unclear to me, whether all fires for the PR2 simulations are estimated on the same Hp or are estimated per individual fire (as described in paragraph 3.4 page 9)? I believe that the magnitude of fires differs and results in different Hp which are influencing the transport of air particles. The explanation could be improved."*

With this line, you can see that the Hp value is calculated for each fire.
"The calculation of Sofiev et al. (2012) is based on the Convective Available Potential Energy estimation, itself diagnosed using the Fire Radiative Power (FRP) of each fire."
and:
"Hp is estimated, for each individual fire, as"

Laurent Menut
November 6, 2017.

**Additional references:**

Marécal V., V.-H. Peuch, C. Andersson, S. Andersson, J. Arteta, M. Beekmann, A. Benedictow, R. Bergstrom, B . Bessagnet, A. Cansado, F. Chéroux, A. Colette, A. Coman, R. L. Curier, H. A. C. Denier van der Gon, A. Drouin, H. Elbern, E. Emili, R. J. Engelen, H. J. Eskes, G. Foret, E. Friese, M. Gauss, C. Giannaros, M. Joly, E. Jaumouillé, B. Josse, N. Kadygrov, J. W. Kaiser, K. Krajsek, J. Kuenen, U. Kumar, N. Liora, E. Lopez, L. Malherbe, I. Martinez, D. Melas, F. Meleux, L. Menut, P. Moinat, T. Morales, J. Parmentier, A. Piacentini, M. Plu, A. Poupkou, S. Queguiner, L. Robertson, L. Rouil, M. Schaap, A. Segers, M. Sofiev, M. Thomas, R. Timmermans, A. Valdebenito, P. van Velthoven, R. van Versendaal, J. Vira, and A. Ung, A regional air quality forecasting system over Europe: the MACC-II daily ensemble production, Geosci. Model Dev., 8, 2777-2813, 2015, www.geosci-model-dev.net/8/2777/2015/, doi:10.5194/gmd-8-2777-2015

Rea G, S.Turquety, L.Menut, R.Briant, S.Mailler and G.Siour, 2015, Source contributions to summertime aerosols in the Euro-Mediterranean region, Atmos. Chem. Phys. Discuss., 15, 8191-8242, 2015, Atmos. Chem. Phys., 15, 8013-8036, doi:10.5194/acp-15-8013-2015

Menut L., G.Rea, S.Mailler, D.Khvorostyanov, S.Turquety, 2015, Aerosol forecast over the Mediterranean area during July 2013 (ADRIMED/CHARMEX), Atmos. Chem. Phys., 15, 7897-7911

Menut L., S.Mailler, G.Siour, B.Bessagnet, S.Turquety, G.Rea, R.Briant, M.Mallet, J.Sciare and P.Formenti, 2015, Ozone and aerosols tropospheric concentrations variability analyzed using the ADRIMED measurements and the WRF-CHIMERE models, Atmos. Chem. Phys., 15, 6159-6182, doi:10.5194/acp-15-6159-2015

Mallet, M., Dulac, F., Formenti, P., Nabat, P., Sciare, J., Roberts, G., Pelon, J., Ancellet, G., Tanré, D., Parol, F., Denjean, C., Brogniez, G., di Sarra, A., Alados-Arboledas, L., Arndt, J., Auriol, F., Blarel, L., Bourrianne, T., Chazette, P., Chevaillier, S., Claeys, M., D'Anna, B., Derimian, Y., Desboeufs, K., Di Iorio, T., Doussin, J.-F., Durand, P., Féron, A., Freney, E., Gaimoz, C., Goloub, P., Gomez-Amo, J. L., Granados-Munoz, M. J., Grand, N., Hamonou, E., Jankowiak, I., Jeannot, M., Léon, J.-F., Maillé, M., Mailler, S., Meloni, D., Menut, L., Momboisse, G., Nicolas, J., Podvin, T., Pont, V., Rea, G., Renard, J.-B., Roblou, L., Schepanski, K., Schwarzenboeck, A., Sellegri, K., Sicard, M., Solmon, F., Somot, S., Torres, B, Totems, J., Triquet, S., Verdier, N., Verwaerde, C., Waquet, F., Wenger, J., and Zapf, P.: Overview of the Chemistry-Aerosol Mediterranean Experiment/Aerosol Direct Radiative Forcing on the Mediterranean Climate (ChArMEx/ADRIMED) summer 2013 campaign, Atmos. Chem. Phys., 16, 455-504, doi:10.5194/acp-16-455-2016, 2016

Mailler S., L. Menut, A. G. Di Sarra, S. Becagli, T. Di Iorio, B. Bessagnet, R. Briant, P. Formenti, J.F. Doussin, J. L. Gomez-Amo, M. Mallet, G. Rea, G. Siour, D. M. Sferlazzo, R. Traversi, R. Udisti, and S. Turquety, 2015, On the radiative impact of aerosols on photolysis rates: comparison of simulations and observations in the Lampedusa island during the CharMEx/ADRIMED campaign, Atmos. Chem. Phys., 16, 1219-1244, doi:10.5194/acp-16-1219-2016

Menut L., G.Siour, S.Mailler, F.Couvidat and B.Bessagnet, 2016, Observations and regional modeling of aerosol optical proprerties, speciation and size distribution over Northern Africa and western Europe, Atmos. Chem. Phys., 6, 12961-12982

Vivanco Marta G., Bertrand Bessagnet; Kees Cuvelier; Mark R Theobald; Svetlana Tsyro; Armin Aulinger; Johannes Bieser; Giuseppe Calori; Giancarlo Ciarelli; Astrid Manders; Mihaela Mircea; Sebnem Aksoyoglu; Gino Briganti; Augustin Colette; Florian COUVIDAT; Andrea Cappelletti; Massimo D'Isidoro; Richard Kranenburg; Frederik Meleux; Laurent Menut; Maria Teresa Pay; Guido Pirovano; Laurence Rouil; Camilo Silibello; Philippe Thunis; Anthony Ung, Joint analysis of deposition fluxes and atmospheric concentrations predicted by six chemistry transport models in the frame of the EURODELTAIII project, 2017, Atmospheric Environment, 151, pp.152-175, http://dx.doi.org/10.1016/j.atmosenv.2016.11.042

---

## Short Comment (SC3) · 7 Nov 2017

**Comment on "Impact of biomass burning on pollutants surface concentrations in mega cities of the Gulf of Guinea" by Laurent Menut et al.**

*Vicky Meulenberg*

*"This review was prepared as part of graduate program course work at Wageningen University, and has been produced under supervision of Prof Wouter Peters. The review has been posted because of its good quality, and likely usefulness to the authors and editor. This review was not solicited by the journal."*

Your manuscript investigates the relative contribution of pollutants caused by biomass burning from central and southern Africa on the surface concentrations of aerosols, carbon monoxide and ozone in urban areas in the Guinean Gulf. For this purpose, a large area is modelled using the Weather and Research Forecast model (WRF) and CHIMERE model. Four simulations were done in the months June and July 2014. The first simulation included the releasing of tracers into the atmosphere to see which regions in central Africa are important for the biomass burning influence in the Gulf of Guinea. It turned out that meteorological conditions are favourable for transporting emissions towards the Gulf cities within one week. The other three simulations investigated the atmospheric content with the CHIMERE model, without biomass burning and with biomass burning injected at two different heights into the troposphere. The simulations were validated with the help of observations and products of the MODIS AOD, AERONET, CO and CALIOP. With the last three simulations the effect of the biomass burning on the total emission concentrations could be investigated and quantified. The modelled results showed no effect of different injection heights far from the sources. Furthermore, the effect of biomass burning appeared after a few days and the maximum contribution of the emissions was for CO 150 µg/m3, for $O_3$ 20 µg/m3 and for $PM_{10}$ 5 µg/m3.

Your manuscript is important because the particle concentrations are rapidly growing in the past couple of years in southern West Africa and at the moment still barely monitored. Whereas, Mari et al. (2008) showed that even biomass burning plumes from the southern hemisphere could reach the Guinean Gulf. It is therefore crucial to be able to quantify the contribution of several processes to the total particle concentration, including biomass burning. Your research is also new and innovative in a sense that the investigated area is to this extend never modelled before. The used methodology fits nicely in the range of subjects of the Atmospheric Chemistry and Physics journal, because a big part of the manuscript is about atmospheric particles and how to model these and one of the main subject areas of the journal is atmospheric modelling.

In general the manuscript is well written, clear and has a good structure. It is directly clear from the goal what the research question is of the manuscript. The calculations schemes and models (such as the Alfaro and Gomes scheme) are thoroughly investigated and improved before you used them in the manuscript in order to answer the research question. Furthermore, The biggest and most important sections end with a small summary. This makes your manuscript very clear and easy to read. If you would not have done this, the paper would be quite long and complicated, but thanks to these summaries, it is easier to grasp immediately the general idea of the manuscript when you read it for the first time. On top of that, you validated the model with all possible satellite observations

and ground measurements that were available. Because almost all the results showed that the model performed well, the model seems very trustworthy. It is therefore in my opinion no problem that the model is not validated in predicting the composition of the pollutants. However, there are some sections and topics, with regard to the research question answer, definition clarity, the chosen research period, errors and some other more minor things, that should be revised in order to make the manuscript truly publishable.

**Major concerns**

My first concern is about the research question. As already said, from the goal it is directly clear what the research question is. In short: investigating the relative contribution of certain pollutants on the surface concentrations in urbanized areas. In your conclusions section you go nicely back to his goal and try to answer this question, but in my opinion you do not completely answer the research question. In short you answer this question by summing up the absolute maximum concentration additions of biomass burning to surface concentrations. This is very nice and interesting but in your goal you promised to give the relative contribution and not the absolute one. This is not only missing in your conclusions section, but you also do not mention it in the results section.

The first question that should be answered is relative to what you want to compare the biomass burning contribution? Relative to the total local air pollution concentrations per substance in the air? Or the contribution of biomass burning per substance to the total concentration of all the substances together emitted by biomass burning? For example Piketh et al. (1999) determines the relative contribution of biomass burning to the total inorganic aerosol concentration. Because you primarily investigate the contribution of the inorganic substances $O_3$ and CO, you could also use this approach. However, because you focus also on the $PM_{10}$, it is in your case more interesting to give the relative contribution of a substance ($O_3$, $PM_{10}$ or CO) emitted by biomass burning to the total substance concentration ($O_3$, $PM_{10}$ or CO) in a certain location.

It is not a lot of work to get the outcome of the relative contribution in your manuscript. Figure 14 shows that you have calculated emissions per substance for FIRES and NoFIRE. If you want the maximum relative contribution, this means that you can easily perform the following calculation for the hour where (FIRES – NoFIRE) is the highest: (FIRES - NoFIRE) / FIRES * 100%, according to Ott et al. (2013). Thus, please state in your introduction relative to what you calculate the biomass burning contribution, calculate in section 7.1 apart from the absolute contributions also the relative contributions with the formula given above and show this result also in the conclusions section. This would make your manuscript better, because you answer the research question properly and a relative contribution provides more information than just the maximum contribution.

The second thing that concerns me is the chosen model period. From the manuscript it becomes clear that the simulations are performed for the period May to July 2014. Reasons to choose this period are that the simulations were a preparation for the fieldwork in June/July 2016 and the onset of the West African Monsoon (WAM) in that period. However, I do not understand why you did not choose for June-August instead of May-July. Several sources (Mari et al. (2008), Williams et al. (2010)) say that the highest concentrations measured due to biomass burning are during August. On top of that, according to Williams et al. (2010) the WAM is from June to August. Furthermore, for the

southern hemisphere, where the tracers are released, the biomass burning season is from June to August (Mari et al (2008)).

For the fieldwork and the tracers you could have modelled August, because if you chose June to August, the model results could still be used for the fieldwork June/July. In the manuscript, you start with releasing the tracer only on 15 June and most of the figures you show are the concentrations at the end of the period: the end of July, because you claim that the concentrations are then the highest. Why are you so sure that concentrations are then the highest?

It would really strengthen the manuscript if you could model the atmospheric composition for August as well in chapter 7, because especially the graphs in figure 14 shows still a positive trend at the end of July. This means that the biomass burning contribution to several substances may be higher in August than at the end of your modelled period. Thus, your estimated maximum contribution values per substances are also too low. These estimated values are very important, because this is the answer to your research question and this answer would not make sense, if the maximum estimated values are in reality much higher a month later. In my opinion it is not necessary to validate the model again for August before you use the model to calculate the atmospheric composition for that month, because figure 6, 10, 11 and 12 show no evidence to assume that the model will perform worse after July 31.

Another concern is that the introduction starts with "*The concentrations of gases and particles are rapidly growing in southern West Africa (SWA), driven by the constant increase of anthropogenic atmospheric emissions".* Whereas, you are going to investigate biomass burning which is a partly naturally caused phenomenon. There are two things in these statements that are not clear for me. First of all, why do you start your introduction with a problem that you do not tackle directly. If you start your introduction like this, I would expect that you try to quantify anthropogenic emission sources for example. This could have been the case if you focussed only on human induced fires, such as agricultural fires and deforestation, which occur in Central Africa (Buccini et al. (2002) , van der Werf et al. (2010)). However with the sentence *"In addition to this anthropogenic regional pollution, the region is impacted by other important sources especially in the summer, with high emissions of mineral dust from the Sahara and Sahel to the north and vegetation fires from Central and southern Africa.",* you implicate that you focus on the natural occurrence of biomass burning. Please make in the introduction clear whether you focus only on anthropogenic induced biomass burning or on naturally induced biomass burning and if you focus only on the latter one, you should make clear in the introduction why we have to investigate biomass burning now, while anthropogenic emissions are growing.

I am also a bit concerned about your (lack of) error propagation in the manuscript. When choosing the values for several model parameters, several calculation schemes and models are used. There are also errors in these, but it is not possible to compare these with the real values, because they are often unknown. It is still very important to be transparent about uncertainties in this stage. For example in the APIFLAME model to calculate the emission fluxes of biomass burning. There are uncertainties in using this model, that you do not mention, such as the fact that Turquety et al. (2014) states that a lot of parameters, such as information about the biomass density are primarily developed for Europe and are more uncertain for the rest of the world (including the Gulf of Guinea). Please be transparent about these kind of uncertainties.

In the next step when you have your model with its parameters, you test the model thoroughly for several aspects and most of the time you conclude that the model is sufficient. However, the correlations between model and observations as shown in table 2 and 3 are on average quite low and often not even 0.3. In my opinion that is too low for a correlation. You tackle this problem for both tables in your text and explain what the reason is for this low correlation (wrong location measurement stations or large temporal variability for example) and why the model is still good enough to use. Still there are errors in the model and in this stage you are able to quantify them. You could do this quite simple and straightforward by calculating an average error percentage per location: divide the showed bias value by the model value. Average these for all the available model-observation comparisons. In the end you have a bias percentage per location for the model.

With the described simple method to determine the bias percentage per location you can include uncertainty bands in your final answers of the conclusions section, even if you did not compare these values with the observations. Now, you just give the averaged values as if it is a fact, but if I regard the previous uncertainty in the modelling which the text clearly explained, there must be uncertainty in these atmospheric composition answers as well. Therefore, you should show the values in the form of CO ≈ 150 ± … µg/m3.

**Minor concerns**

*General structure remark:* Until figure 7, the figures are not closely placed to the text where they are explained. Please put the figures close to their text part.

*Page 1, line 16:* There is no reference after this sentence or small section. There are several sources which say that southern West Africa has a big air pollution problem (Knippertz et al. (2015), Liousse et al. (2014) and De Longueville et al. (2010)), but I could not find a source that states that especially the mentioned areas have the biggest problems. The second part of the sentence about the atmospheric boundary layer is from Real et al. (2010), which is mentioned some sentences later. Can you provide the correct references after this sentence?

*Page 1, Abstract/Introduction general:* Page 15 to 25 of the manuscript, which is almost 1/3 of the total content, is about comparing the model runs with observations. Thus, the model is validated on several levels. It is very good that the model is thoroughly tested, before using it without observations, but from the introduction and abstract it is not clear that such a big part of the manuscript is about validating. There is only one sentence that states shortly that the meteorological model ability and chemical species concentration prediction ability will be compared with observations. Please state clearly in your introduction and abstract that the model is thoroughly validated, with the help of the observations of space-born platforms and ground based stations, such as IASI, AERONET, CALIOP… This fits nicely with the next section which gives a more detailed description of the used observations.

*Page 1, line 18:* You give Real et al. (2010) as a reference. In my opinion Real et al. (2010) only shows that biomass burning pollutants can reach the Gulf area from Central Africa, but says nothing about Sahara sand or how important this source is. Thus, by the given reference I am certainly not convinced. Mari et al. (2008), to which you refer in line 2, states clearly that the region in general is impacted by biomass burning. You should refer to Mari et al. (2008) instead of Real et al. (2010). Still, none of these sources says something about mineral dust. De Longueville et al. (2010) states that

especially in West Africa mineral dust is a very important factor affecting the local air quality, thus you should also refer to De Longueville et al. (2010) instead of Real et al. (2010).

*Page 2, line 6:* If you are as a reader not familiar with the DACCIWA project it is entirely unclear what the link is between the project and this paper. Is the manuscript financially supported by this project? From which organisation is this project? This is now only clear if you read the Acknowledgements in the end.

*Page 4, figure 1:* Figure should be a little bit bigger to make it easier to read. The disadvantage of having the figure already on page 4 is that it gives also an overview of the CALIOP and IASI locations, whereas at that point you do not know what the purpose is of CALIOP and IASI. A sentence about this in the text of the manuscript would be nice.

*Page 6, line 22:* You use updated data about the erodibility provided by Beegum et al (2016). However, you say nothing about using updated data for the roughness length, whereas this information can also be provided by Beegum et al. (2016). It is therefore not clear for me whether you used the updated information about roughness length in the manuscript. Can you make this clear?

*Page 18, table 2:* Several aspects of this table are not completely clear. Can you explain why there is for almost all the locations so little data available and how this does not influence the validation? Furthermore, if you scan the table for the first time it is definitely not clear that 'Obs' and 'Model' are just the real measured or simulated values, because there is nothing about that in the caption and you give no units. Please give units for these values. On top of that, it is not directly clear that Rt is the correlation where you are talking about in the text, why do you call it temporal correlation in the tables? Could you explain this in the text? These last two points apply also to table 3.

*Page 25, section 6.6:* This section concludes that the model performed quite good in all the tests. However some observation-model comparisons showed that there were uncertainties and prediction errors. Can you summarize these model uncertainties as well?

**Minor issues**

*Page 1, line 14:* "linked to" instead of "linked with".

*Page 2, line 27:* "Ground-based" instead of "groud-based".

*Page 6, figure 2:* The purpose of this figure is completely unclear for me. It shows the modelled anthropogenic $NO_2$ fluxes for a week day. It is interesting to see output of the CHIMERE model, but the modelled flux is not investigated in the paper, so why would you show it? In my opinion, this figure does not add value to the manuscript and can be deleted.

*Page 7, line 11:* "Of the daily …" Of the daily what? I think the word "fluxes" is missing.

*Page 8, line 12:* "A homogeneous way" instead of "an homogeneous way".

*Page 9, line 18:* "Consist of" instead of "consist in".

*Page 10, section 4:* This section comes a little bit out of the blue here. In essence you compare the model with the observations. Therefore, I think that this section fits better as a new section 6.1 in section 6. Because in section 6 you compare the model with observations and in the small introduction of section 6 you come back to the results of section 4: it is better to provide this results directly in section 6.

*Page 12, figure 6:* Can be deleted. You do not refer in the text to figure 6 and it is actually unnecessary, because figure 5 shows the same idea, but then on a spatial scale.

*Page 17, line 10:* "Correctly enough" instead of "enough correctly".

*Page 20, figure 11:* This figure shows the comparison between model and observations at different locations. Per location there is a graph for east and west. In my opinion there is barely difference between east and west for every location. It would therefore be better the say in the text that there is no difference between east and west and just show one graph per location (south, north central).

*Page 21, line 13:* "Where the studied areas are located" instead of "where are located the studied areas".

*Page 21, line 14:* "Have" instead of "has".

*Page 22, table 3:* Caption claims that RMSE, the root mean square error, can also be seen in the table. This is not the case. You should delete this from the caption or include indeed the RMSE.

*Page 24, line 27:* "But" instead of "bit".

*Page 28, line 12:* "Period in" instead of "period of".

**References**

Beegum, S. N., Gherboudj, I., Chaouch, N., Couvidat, F., Menut, L., & Ghedira, H. (2016). Simulating aerosols over Arabian Peninsula with CHIMERE: Sensitivity to soil, surface parameters and anthropogenic emission inventories. *Atmospheric Environment*, *128*, 185-197.

Knippertz, P., Coe, H., Chiu, J. C., Evans, M. J., Fink, A. H., Kalthoff, N., ... & Danour, S. (2015). The DACCIWA project: Dynamics–aerosol–chemistry–cloud interactions in West Africa. *Bulletin of the American Meteorological Society*, *96*(9), 1451-1460.

Liousse, C., Assamoi, E., Criqui, P., Granier, C., & Rosset, R. (2014). Explosive growth in African combustion emissions from 2005 to 2030. *Environmental Research Letters*, *9*(3), 035003.

De Longueville, F., Hountondji, Y. C., Henry, S., & Ozer, P. (2010). What do we know about effects of desert dust on air quality and human health in West Africa compared to other regions?. *Science of the Total Environment*, *409*(1), 1-8.

Mari, C. H., Cailley, G., Corre, L., Saunois, M., Attié, J. L., Thouret, V., & Stohl, A. (2008). Tracing biomass burning plumes from the Southern Hemisphere during the AMMA 2006 wet season experiment. *Atmospheric Chemistry and Physics*, *8*(14), 3951-3961.

Ott, R. L., & Longnecker, M. T. (2015). *An introduction to statistical methods and data analysis*. Nelson Education.

Piketh, S. J., Annegarn, H. J., & Tyson, P. D. (1999). Lower tropospheric aerosol loadings over South Africa: The relative contribution of aeolian dust, industrial emissions, and biomass burning. *Journal of Geophysical Research: Atmospheres*, *104*(D1), 1597-1607.

Real, E., Orlandi, E., Law, K. S., Fierli, F., Josset, D., Cairo, F., ... & McQuaid, J. B. (2010). Cross-hemispheric transport of central African biomass burning pollutants: implications for downwind ozone production. *Atmospheric Chemistry and Physics*, *10*(6), 3027-3046.

Turquety, S., Menut, L., Anav, A., Viovy, N., Maignan, F., & Wooster, M. J. (2014). APIFLAME v1. 0: high-resolution fire emission model and application to the Euro-Mediterranean region. *Geoscientific Model Development*, *7*, 587-612.

Van der Werf, G. R., Randerson, J. T., Giglio, L., Collatz, G. J., Mu, M., Kasibhatla, P. S., ... & van Leeuwen, T. T. (2010). Global fire emissions and the contribution of deforestation, savanna, forest, agricultural, and peat fires (1997–2009). *Atmospheric Chemistry and Physics*, *10*(23), 11707-11735.

Williams, J. E., Scheele, M. P., van Velthoven, P. F. J., Thouret, V., Saunois, M., Reeves, C. E., & Cammas, J. P. (2010). The influence of biomass burning and transport on tropospheric composition over the tropical Atlantic Ocean and Equatorial Africa during the West African monsoon in 2006. *Atmospheric Chemistry and Physics*, *10*(20), 9797-9817.

---

## Referee Comment (RC1) · Anonymous Referee #1 · 21 Nov 2017

ACPD doi:10.5194/acp-2017-852

Summary: This paper presents modeling results from the DACCIWA project. Model results are performed for the period May – July 2014, with and without biomass burning emissions. The model results have been compared to a variety of observations. The model simulations appear to be of high quality, and the authors do compare the model to observations and not the weaknesses/strengths of the simulation. However, the paper is tediously long and very un-focused. There are 15 Figures, many of which could be moved to SI. There are also randomly short sections (e.g. Section 7.1 and 7.2 which contain only a few sentences each). Many of the Figures are model output

for specific days, but the logic behind the choice of day is hard to follow. Thus I can't recommend that this paper be accepted for publication in ACP as it currently is formatted. I see that there has been a class exercise devoted to reviewing this article, and they note several grammar issues. I also see quite a few grammar issues, but I have not pointed them out specifically because I think re-structuring is necessary. Here is my recommendation for re-structuring.

1) Begin with a single large map (similar to Figure 1) that shows the locations of the urban areas of interest, and biomass burning emissions during this period of time. Remove unnecessary figure clutter, and label the legend. This is currently not done in Figures 2 and 3. The current versions of Figure 2 and 3 can be omitted or moved to supplemental.

2) Separate the observations from the modeling. It does not makes sense to have sites with no measurements listed in Table 1.

3) Describe the essential components of the modeling in the methods, and move some of this information to SI. This section is currently 5 pages. The documentation is good for reproducibility, but can be moved to SI.

4) Begin the paper by showing the observations, rather than the model results. Figure 9 would be a good place to start. This Figure shows large areas of high AOD corresponding to fire and dust emissions. The legend should be labeled (not with a green highlighted text, but rather next to the legend in plain English). Then I would move to describe Figure 10. It would be good to focus on specific events that are simulated well and those that are not simulated well. Highlight those events with colors. From here, the authors could present Figure 8, which shows the maps of surface tracer concentrations for a specific day (27 July). This Figure should be clearly linked to Figure 10 and potentially to Figure 11 and 12. Without this link, the choice of model output seems very random.

5) Push most of the model validation to the SI.

6) End the discussion by noting the key points of Figure 14 and Figure 15. Please put the gas phase species in ppbv rather than ug/m3. It is unclear if Figure 15 is the amount of PM10 attributed to fires. The caption is confusing.

---

## Referee Comment (RC2) · M. Parrington (Referee) · 28 Nov 2017

Menut et al. present a comprehensive evaluation of the influence of biomass burning emissions in central Africa on atmospheric composition and air quality in cities around the Gulf of Guinea. They clearly show that their model represents the atmospheric state in good agreement with different observational datasets. Their evaluation of the model sensitivity to parameters related to the injection height of biomass burning emissions is very welcome and clearly explained. I recommend publication of this work in Atmospheric Chemistry and Physics subject to addressing the following comments.

General comments

[Figure]

I recommend that the authors check through the manuscript for the consistent use of some expressions. In particular they exchange "Gulf of Guinea" with "Guinean Gulf" quite regularly and it would read more clearly if they chose one and use that. Also the terminology of the different experiments (e.g., NoFIRE and FIRES) is a bit mixed up through the manuscript after it has already been defined – they can use the experiment names without having to describe them again.

The manuscript touches on a couple of key issues in biomass burning and atmospheric composition on which some further comments or recommendations would be useful. Firstly, some concluding comment on injection heights for biomass burning emissions would be of interest, particularly on whether it improves the estimation or not. Secondly, the CHIMERE model underestimates column CO compared with observations which is commonly seen across different models – can the authors make some comment on why this is the case with CHIMERE? In particular the injection height of the fire emissions doesn't seem to make up the difference, could the OH field in the model be playing a role? I don't expect them to answer these questions fully but some comments in the context of the presented work would be a welcome addition.

Specific comments

Page 1, line 7-8: replace "to be" with "being".

Page 2, line 2: replace "have evidenced" with "show".

Page 2, line 27: "groud" should be "ground".

Page 2, line 28: replace "Satellites data provide" with "Satellite data provides".

Page2, line 32: define the AERONET acronym – it is given in the caption for Table 1 but should be included in the text and removed from the Table caption.

Page 4, line 1: the authors mix up use of "modelled" and "modeled" – please check consistency throughout the manuscript (ideally using "modelled").

Page 5, line 19: "consist in" should be "consist of". Also check rest of manuscript.

Page 5, line 27: it isn't clear what is meant by "monthly databases".

Page 5, line 32: the last sentence isn't very clear – are Abidjan and Lagos the only megacities? Or are there more?

Page 6, Figure 2 caption: check the units are consistent with those given in the main text.

Page 6, line 15: replace "were done" with "have been made".

Page6, line 29: replaces "source" with "sources".

Page 7, line 10: "area burned" should be "burned area", "daily estimated" should be "estimated daily".

Page 7, line 11: clarify that daily refers to the daily emission.

Page 7, line 12: "CO for the month".

Page 7, line 14: clarify if "South-Africa" refers to SWA.

Page 8, line 16: "numerically cost consuming".

Page 9, line 11: "fires" should be "fire".

Page 9, line 32: clarify if "vegetation emissions fluxes" refers to biogenic emissions or fire emissions.

Page 9, line 34: use "concentrations" rather than "content".

Page 10, first paragraph: the explanation of the different experiments has already been given in the previous pages – I found the explanation clearer on this page and it would benefit the reader to use just this one, linking to the TRC and FIRES experiments as already described.

Page 10, line 11: "information is". The details of the plots in Figure 1 are not necessary

in the text and should be removed.

Page 10, lines 15 and 17: use "south" rather than "bottom"?

Page 11, Figure 5 caption: not necessary to specify BADC as the source of observations – this is clear in the text.

Page 11, line 10: "fire emissions"?

Page 12, line 1: "dry convection in the lower troposphere".

Page 15, line 1: replace "concentrations" with smoke or pollution?

Page 15, line 2: replace "catched" with "caught".

Page 15, line 11: "continental Central Africa".

Page 15, Section 5.3: it isn't clear that this section is all that necessary as much of the summary of results has already been made in the preceding points.

Page 16, line 1: "MODIS AOD product at wavelength of 500nm" – use of the symbol lambda is unnecessary, also using phi for latitude later in the manuscript.

Page 16, Figure 9 caption: should "CHIMERE without fires" be NoFIRE and "CHIMERE with vegetation fire emissions" be FIRES? Also it would be of great benefit to have each plot of the figure labelled so that a clearer explanation can be given in the caption – this also applies to the other figures.

Page 16, line 11: "fires emissions" should be "fire emissions" – please also change this throughout the manuscript.

Page 16, line 13: "shows that the model".

Page 16, lines 13 and 14: please quantify the terms "underestimated" and "overestimated" – it isn't easy to tell from the Figure, and the addition of difference plots would be useful.

Page 17, line 8: "BADC stations" – please clarify the source of the measurements (Met Office MIDAS land surface stations?) and not the data centre where they were obtained from.

Page 17, line 10: change "this" to "it", and "not modelled enough correctly" to "not modelled correctly".

Page 17, line 14: "biomass burning" (use either fire or biomass burning consistently.

Page 17, line 20: what is meant by "not well retrieved"? does this refer to the satellite observations? Or how the model represents the atmospheric concentrations? "the Central Africa" should be "Central Africa", and "AODs" should be "AOD" (please change this throughout the manuscripts as well).

Page 17, lines 35 and 38: "FIREs" should be "FIRES" or "FIRE" – also throughout the rest of the manuscript.

Page 17, line 37: a comment on how difficult it is to reproduce the long-range transport over the ocean due to errors in the model transport would be useful.

Page 18, line 8: replace "retrieved" with "represented"?

Page 18, line 14-15: suggest changing to "the hypothesis that the optical properties of the modelled aerosols, and the estimation of the extinction".

Page 19, Figure 10 caption: clarify which column is which – in the text it is described but should also be here in the caption. Labelling the plots will be helpful.

Page 19, line 6: "expressed" should be "expresses".

Page 19, line 8: "while high values".

Page 19, line 11: change "finest particules" to "finer particles".

Page 20, Figure 11 caption: "carbon monoxide", no need to redefine the different experiments (i.e., remove "without" and "and with biomass burning").

[Figure]

Page 20, line 6: "consists of three-day averaged".

Page 21, line 4-5: the description of the three model simulation types is not necessary.

Page 21, line 8: clarify that these are CO columns rather than concentrations.

Page 20, line 13: change "where are located the studied cities" to "where the studied cities are located".

Page 20, line 14: "has" should be "have".

Page 20, section 6.4, line 6: "colocated" should be "collocated".

Page 20, section 6.4, line 16: remove "correctly" – as described in the manuscript, the model does not 100% correspond to the observations.

Page 21, line 1: the first sentence isn't very clear – is the author referring to modelling the vertical aerosol profile for comparison against CALIOP? If this is the case it can be stated more simply.

Page 23, line 1: "thresholds of optical characteristics"?

Page 23, line 4: "not being modelled".

Page 24, line 24: change "retrieved" to "reproduced".

Page 24, line 27: "bit" should be "but".

Page 25, line 11: "enables us to have".

Page 25, line 15: "For the fire emissions".

Page 25, line 25: "tree" should be "three"; change "both produced by" to "produced by both".

Page 25, section 7.1, line 6: "no pollution peaks".

Page 26, line 1: "a pollution alert"?

[Figure]

Page 26, Figure 14: it would be useful, in the context of pollution alerts to relate the reported CO, O3 and PM10 results to the WHO and/or EU recommended exposure thresholds. Add plot labels.

Section 7.2: This section would be of great interest if there were any corroborating observations, which does not seem to have been the case here, or if the authors could provide references to where aerosol composition from the model has been compared against observations. In its current form this section seems to be a bit of an unnecessary addition and distracts from the main message of the rest of the manuscript and could be removed.

Page 28, line 12: change "pollutants surface concentrations" to "surface concentrations of pollutants".

---

## Author Response (AR1)

**Impact of biomass burning on pollutants surface concentrations in megacities of the Gulf of Guinea**

Laurent MENUT, Cyrille FLAMANT, Solène TURQUETY,
Adrien DEROUBAIX, Patrick CHAZETTE and Rémi MEYNADIER

Dear Editor and reviewers,

We acknowledge the reviewers for the time spent to evaluate our work. We also acknowledge the Editor and we made all proposed changes in the revised manuscript. Please note that our answers are in blue in the text and after each reviewer's remark.

Best regards,
Laurent MENUT
December 21, 2017

**1 Reviewer #1**

**Summary:**

This paper presents modeling results from the DACCIWA project. Model results are performed for the period May - July 2014, with and without biomass burning emissions. The model results have been compared to a variety of observations. The model simulations appear to be of high quality, and the authors do compare the model to observations and not the weaknesses/strengths of the simulation.

Paragraphs were added in the revised version to better describe the weaknesses/strengths of the simulation.

However, the paper is tediously long and very un-focused. There are 15 Figures, many of which could be moved to SI. There are also randomly short sections (e.g. Section 7.1 and 7.2 which contain only a few sentences each). Many of the Figures are model output for specific days, but the logic behind the choice of day is hard to follow.

The article was reorganized and the English was completely checked. Some material was moved in several Appendices. Some subsections were reorganized to have more homogeneous sections. Text was also added to have better explanations about the choice of the days presented as examples. The selected days correspond to: (i) the end of the modelled period and (ii) the days where CALIOP data were available and interesting in term of biomass burning plumes. This was also added in the text.

Thus I can't recommend that this paper be accepted for publication in ACP as it currently is formatted. I see that there has been a class exercise devoted to reviewing this article, and they note several grammar issues.

Three comments were received during the discussion phase of this paper. They were all posted by students from the same class. They considered that it was an "exercise of review". But they are not reviewers. The ACP principle is to have (i) scientific reviews from professional, selected by the editor, or (ii) short comments. The short comments have to be "short" and "comments". The goal is to promote a dialog and improve the paper. And it was not the case with these comments. A long answer was written for the first one. The others were considered as not constructive and after discussion with the Editor, it was decided not to answer the second and third comments.

I also see quite a few grammar issues, but I have not pointed them out specifically because I think re-structuring is necessary. Here is my recommendation for re-structuring.

We acknowledge the Reviewer for these remarks. Following his/her recommendations, we simplified the article, with material in several Appendices, less figures in the main text and a Section devoted to the presentation of observations and a Section focusing on new model developments only .

**Recommendations:**

1. Begin with a single large map (similar to Figure 1) that shows the locations of the urban areas of interest, and biomass burning emissions during this period of time. Remove unnecessary figure clutter, and label the legend. This is currently not done in Figures 2 and 3. The current versions of Figure 2 and 3 can be omitted or moved to supplemental.

   The Figures for the emissions are now in an Appendix, with the corresponding description of the model.

2. Separate the observations from the modeling. It does not makes sense to have sites with no measurements listed in Table 1.

   We moved the description of the sites of interest that did not reflect actual measuring sites to the Table in the 'tracers' section.

3. Describe the essential components of the modeling in the methods, and move some of this information to SI. This section is currently 5 pages. The documentation is good for reproducibility, but can be moved to SI.

   A large part of the model description is now in a Appendix.

4. Begin the paper by showing the observations, rather than the model results. Figure 9 would be a good place to start. This Figure shows large areas of high AOD corresponding to fire and dust emissions. The legend should be labeled (not with a green highlighted text, but rather next to the legend in plain English). Then I would move to describe Figure 10. It would be good to focus on specific events that are simulated well and those that are not simulated well. Highlight those events with colors. From here, the authors could present Figure 8, which shows the maps of surface tracer concentrations for a specific day (27 July). This Figure should be clearly linked to Figure 10 and potentially to Figure 11 and 12. Without this link, the choice of model output seems very random.

   There was a logic to present the results as it was. First the meteorology, then the tracers, then the full simulation. CALIPSO data can be compared only at the end, after the full simulation. Nevertheless, we have decided to comply with the referees suggestion and we have restructured the paper accordingly.

5. Push most of the model validation to the SI.

   Many figures and text are now in an Appendix.

6. End the discussion by noting the key points of Figure 14 and Figure 15. Please put the gas phase species in ppbv rather than ug/m3. It is unclear if Figure 15 is the amount of PM10 attributed to fires. The caption is confusing.

The caption was corrected. For the units, it is better to have concentrations in $\mu$g m$^{-3}$. ppbv is a unit mainly used in climate modelling, and related to the upper troposphere and lower stratosphere, since this is a unit designed to expressed very low quantities, with no dimension and independent on the air density (even though dependent on the altitude: 5 ppbv at altitude 5km is not the same amount of pollutants than 5 ppbv at 10km). For atmospheric pollution, as in this study, and in the boundary layer, the unit is $\mu$g m$^{-3}$. This unit is the one for all surface stations and of many research papers (see atmospheric composition papers in ACP or GMD).

Many parts of this paper were changed or moved at the end of the manuscript in several Appendices. We hope that, now, the structure of the paper and content of the paper are clearer to the reviewer.

**2  Reviewer #2**

Menut et al. present a comprehensive evaluation of the influence of biomass burning emissions in central Africa on atmospheric composition and air quality in cities around the Gulf of Guinea. They clearly show that their model represents the atmospheric state in good agreement with different observational datasets. Their evaluation of the model sensitivity to parameters related to the injection height of biomass burning emissions is very welcome and clearly explained. I recommend publication of this work in Atmospheric Chemistry and Physics subject to addressing the following comments.

Thanks a lot for these positive comments.

**General comments**

I recommend that the authors check through the manuscript for the consistent use of some expressions. In particular they exchange "Gulf of Guinea" with "Guinean Gulf" quite regularly and it would read more clearly if they chose one and use that. Also the terminology of the different experiments (e.g., NoFIRE and FIRES) is a bit mixed up through the manuscript after it has already been defined - they can use the experiment names without having to describe them again.

The acronyms and grammar was completely checked and corrected. For example, this is now "Gulf of Guinea" in the whole manuscript. A complete check was also done for the simulations names.

The manuscript touches on a couple of key issues in biomass burning and atmospheric composition on which some further comments or recommendations would be useful. Firstly, some concluding comment on injection heights for biomass burning emissions would be of interest, particularly on whether it improves the estimation or not.

About this point, the following paragraph was added in the revised version and in the conclusion: "In order to reduce the uncertainty in the simulation due to the way to inject biomass burning emissions in the atmosphere, two simulations were performed with different vertical profiles. It was shown that modelled results were not sensitive to the shape of the profile. The reason is that, during a fire, the pyroconvection induces a strong and fast mixing of the surface flux. Whatever the shape of the injection profile, the pollutants are finally quickly vertically mixed before a long-range transport."

Secondly, the CHIMERE model underestimates column CO compared with observations which is commonly seen across different models - can the authors make some comment on why this is the case

with CHIMERE? In particular the injection height of the fire emissions doesn't seem to make up the difference, could the OH field in the model be playing a role?

*We agree this is an important question. In fact, we tested the injection profile shape and not the injection height. The injection height is parameterized fire per fire and using the [Sofiev et al., 2012] scheme. We consider this scheme to be validated and we used it without changes. About the CO underestimation, the following paragraph was added in the revised version: The differences between observations and model may be due to several factors: First, the boundary conditions used for the simulations are global and 'climatological' in model outputs. The transition from 'mean' values and this real test case may induce biases. For long-lived species such as CO, these biases may be transported inside the model domain. Secondly, underestimated CO may be due to overestimated OH or to an underestimate of the production of CO from the oxidation of VOCs. [Zeng et al., 2015] showed that this last process results in a large variability in model results. However, without complementary observations it remains difficult to disentangle different contributions.*

I don't expect them to answer these questions fully but some comments in the context of the presented work would be a welcome addition.

*All corrections presented below were done.*

**Specific comments**

- Page 1, line 7-8: replace "to be" with "being".

  *OK corrected.*

- Page 2, line 2: replace "have evidenced" with "show".

  *OK corrected.*

- Page 2, line 27: "groud" should be "ground".

  *OK corrected.*

- Page 2, line 28: replace "Satellites data provide" with "Satellite data provides".

  *OK corrected.*

- Page2, line 32: define the AERONET acronym - it is given in the caption for Table 1 but should be included in the text and removed from the Table caption.

  *OK corrected.*

- Page 4, line 1: the authors mix up use of "modelled" and "modeled" - please check consistency throughout the manuscript (ideally using "modelled").

  *"Modelled" is the British spelling, "modeled" is the US spelling. So, it seems that the ideal is British. We agree and the word was changed accordingly.*

- Page 5, line 19: "consist in" should be "consist of". Also check rest of manuscript.

  *OK done (three times).*

- Page 5, line 27: it isn't clear what is meant by "monthly databases".

  *It means that HTAP propose maps of anthropogenic emissions mass on the basis of one month. Thus, there is 12 global maps available, one per month. Depending on the modelled period, we are using the maps corresponding to the month we are modelling. All this part is now in a Appendix, following the request of Reviewer #1. And more explanations were added in this Appendix about these emissions.*

- Page 5, line 32: the last sentence isn't very clear - are Abidjan and Lagos the only megacities? Or are there more?

  The sentence was corrected. Abidjan and Lagos are two megacities, there is others along the coast.

- Page 6, Figure 2 caption: check the units are consistent with those given in the main text.

  There was an error in the caption and it was corrected. This is ($g\ m^{-2}\ day^{-1}$).

- Page 6, line 15: replace "were done" with "have been made".
- Page6, line 29: replaces "source" with "sources".

  OK corrected.

- Page 7, line 10: "area burned" should be "burned area", "daily estimated" should be "estimated daily".

  OK corrected.

- Page 7, line 11: clarify that daily refers to the daily emission.

  OK corrected.

- Page 7, line 12: "CO for the month".

  OK corrected.

- Page 7, line 14: clarify if "South-Africa" refers to SWA.

  Yes this is SWA and it is corrected.

- Page 8, line 16: "numerically cost consuming".

  OK corrected.

- Page 9, line 11: "fires" should be "fire".

  OK corrected.

- Page 9, line 32: clarify if "vegetation emissions fluxes" refers to biogenic emissions or fire emissions.

  OK corrected. This is 'biomass burning emissions'

- Page 9, line 34: use "concentrations" rather than "content".

  OK corrected.

- Page 10, first paragraph: the explanation of the different experiments has already been given in the previous pages - I found the explanation clearer on this page and it would benefit the reader to use just this one, linking to the TRC and FIRES experiments as already described.

  OK the paragraph was merged with the explanations on the previous pages.

- Page 10, line 11: "information is". The details of the plots in Figure 1 are not necessary in the text and should be removed.

  OK, it is right, this part is already in the caption and was removed.

- Page 10, lines 15 and 17: use "south" rather than "bottom"?

  OK corrected.

- Page 11, Figure 5 caption: not necessary to specify BADC as the source of observations - this is clear in the text.

  OK corrected.

- Page 11, line 10: "fire emissions"?

  OK corrected. This is 'biomass burning emissions'.

- Page 12, line 1: "dry convection in the lower troposphere".

  OK corrected.

- Page 15, line 1: replace "concentrations" with smoke or pollution?

  Here, this is the tracers experiment. This is not adapted to add smoke or pollution. This is just concentrations of a tracer.

- Page 15, line 2: replace "catched" with "caught".

  OK corrected.

- Page 15, line 11: "continental Central Africa".

  OK corrected.

- Page 15, Section 5.3: it isn't clear that this section is all that necessary as much of the summary of results has already been made in the preceding points.

  This is not mandatory and we suppressed these intermediary conclusions.

- Page 16, line 1: "MODIS AOD product at wavelength of 500nm" - use of the symbol lambda is unnecessary, also using phi for latitude later in the manuscript.

  OK corrected.

- Page 16, Figure 9 caption: should "CHIMERE without fires" be NoFIRE and "CHIMERE with vegetation fire emissions" be FIRES? Also it would be of great benefit to have each plot of the figure labelled so that a clearer explanation can be given in the caption - this also applies to the other figures.

  OK titles were added in the Figure.

- Page 16, line 11: "fires emissions" should be "fire emissions" - please also change this throughout the manuscript.

  OK corrected.

- Page 16, line 13: "shows that the model".

  OK corrected.

- Page 16, lines 13 and 14: please quantify the terms "underestimated" and "over- estimated" - it isn't easy to tell from the Figure, and the addition of difference plots would be useful.

  A quantification of the differences was added in the text. The difference plots would add a lot of Figures, when the Reviewer #1 considers there is already too many Figures. Here the only message is about the most important differences between the observations and the three simulations. For that, a quantification in the text is enough.

- Page 17, line 8: "BADC stations" - please clarify the source of the measurements (Met Office MIDAS land surface stations?) and not the data centre where they were obtained from.

  Yes, this is "Met Office MIDAS land surface stations" as written. The sentence was simplified.

- Page 17, line 10: change "this" to "it", and "not modelled enough correctly" to "not modelled correctly".

  OK corrected.

- Page 17, line 14: "biomass burning" (use either fire or biomass burning consistently.

  The word "fires" was replaced by FIRE because we were talking about the simulation in this case.

- Page 17, line 20: what is meant by "not well retrieved"? does this refer to the satellite observations? Or how the model represents the atmospheric concentrations? "the Central Africa" should be "Central Africa", and "AODs" should be "AOD" (please change this throughout the manuscripts as well).

  OK corrected.

- Page 17, lines 35 and 38: "FIREs" should be "FIRES" or "FIRE" - also throughout the rest of the manuscript.

  OK corrected throughout the rest of the manuscript, including the Figures.

- Page 17, line 37: a comment on how difficult it is to reproduce the long-range transport over the ocean due to errors in the model transport would be useful.

  The following sentence was added in the manuscript: "The low score in Ascension is related to the location of the site and the fact that the long range transport over the sea is difficult to reproduce: being less turbulent, there is less horizontal diffusion and vertical mixing. The plumes are thinner and more concentrated and the results are more sensitive to a possible model error on the wind direction. The comparison to observations located at one single point over the sea is thus often less correlated than comparisons over land."

- Page 18, line 8: replace "retrieved" with "represented"?

  Yes, sure. This was done.

- Page 18, line 14-15: suggest changing to "the hypothesis that the optical properties of the modelled aerosols, and the estimation of the extinction".

  OK, corrected.

- Page 19, Figure 10 caption: clarify which column is which - in the text it is described but should also be here in the caption. Labelling the plots will be helpful.

  OK, the caption is now: "Comparison of AERONET measurements and model results for the AOD (left) and Angstrom exponent (right). Time series are presented for the Cinzana and Lope stations and for the whole modelled period."

- Page 19, line 6: "expressed" should be "expresses".

  OK, corrected.

- Page 19, line 8: "while high values".

  OK, corrected.

- Page 19, line 11: change "finest particules" to "finer particules".

  OK, corrected.

- Page 20, Figure 11 caption: "carbon monoxide", no need to redefine the different experiments (i.e., remove "without" and "and with biomass burning").

  OK, corrected.

- Page 20, line 6: "consists of three-day averaged".

  OK, corrected.

- Page 21, line 4-5: the description of the three model simulation types is not necessary.

  OK, corrected.

- Page 21, line 8: clarify that these are CO columns rather than concentrations.

  The new sentence is now: "The IASI data show the increase of vertically integrated CO concentrations over Central Africa..."

- Page 20, line 13: change "where are located the studied cities" to "where the studied cities are located".

  OK, corrected.

- Page 20, line 14: "has" should be "have".

  OK, corrected.

- Page 20, section 6.4, line 6: "colocated" should be "collocated".

  OK, corrected.

- Page 20, section 6.4, line 16: remove "correctly" - as described in the manuscript, the model does not 100% correspond to the observations.

  OK, corrected.

- Page 21, line 1: the first sentence isn't very clear - is the author referring to modelling the vertical aerosol profile for comparison against CALIOP? If this is the case it can be stated more simply.

  The new sentence is: "The CALIOP lidar measurements, on-board the CALIPSO satellite [Winker et al., 2010], are analyzed to obtain an aerosol sub-type classification (CALIOP v4.10 product), as proposed in [Omar et al., 2010] and [Burton et al., 2015]."

- Page 23, line 1: "thresholds of optical characteristics"?

  OK, corrected.

- Page 23, line 4: "not being modelled".

  OK, corrected.

- Page 24, line 24: change "retrieved" to "reproduced".

  OK, corrected.

- Page 24, line 27: "bit" should be "but".

  OK, corrected.

- Page 25, line 11: "enables us to have".

  OK, corrected.

- Page 25, line 15: "For the fire emissions".

  OK, corrected.

- Page 25, line 25: "tree" should be "three"; change "both produced by" to "produced by both".

  OK, corrected.

- Page 25, section 7.1, line 6: "no pollution peaks".

  OK, corrected.

- Page 26, line 1: "a pollution alert"?

  A pollution alert is when the concentrations is larger than predefined thresholds. But, in the context of this study, this is not mandatory to speak about that and it was removed.

- Page 26, Figure 14: it would be useful, in the context of pollution alerts to relate the reported CO, O3 and PM10 results to the WHO and/or EU recommended exposure thresholds. Add plot labels.

  The plot labels are already present and correspond to the location (top), the aerosol type (legend) and the plotted variables (left). The abscissa corresponds to the time, but it is mentioned "Time series" in the caption, and the x-axis is explicitly with days. The text about 'pollution alerts' was removed here, because the conversion of 'pollutants concentrations' and 'threshold and exposure' is a particular context and devotes probably a specific study.

- Section 7.2: This section would be of great interest if there were any corroborating observations, which does not seem to have been the case here, or if the authors could provide references to where aerosol composition from the model has been compared against observations. In its current form this section seems to be a bit of an unnecessary addition and distracts from the main message of the rest of the manuscript and could be removed.

  We think this section is of interest because there is no measurement. In general, observations for surface concentrations are limited to $PM_{2.5}$ and $PM_{10}$. Some rare measurements are done for chemical species in Europe, but this is very limited and mainly done during specific neasurements campaign. This is also rare to have the information with the models, because it is necessary to have a chemistry-transport model integrating all aerosol types, thus all sources and chemistry. This is the case with the CHIMERE model and this is why we consider this Figure is an interesting added value for this study, since we want to quantify how much and what aerosol are involved in the biomass burning emissions. This aerosol composition was already studied with CHIMERE in [Menut et al., 2016] and this reference was added in the text.

- Page 28, line 12: change "pollutants surface concentrations" to "surface concentrations of pollutants".

  OK, corrected.

**References**

[Burton et al., 2015] Burton, S. P., Hair, J. W., Kahnert, M., Ferrare, R. A., Hostetler, C. A., Cook, A. L., Harper, D. B., Berkoff, T. A., Seaman, S. T., Collins, J. E., Fenn, M. A., and Rogers, R. R. (2015). Observations of the spectral dependence of linear particle depolarization ratio of aerosols using nasa langley airborne high spectral resolution lidar. *Atmospheric Chemistry and Physics*, 15(23):13453–13473.

[Menut et al., 2016] Menut, L., Siour, G., Mailler, S., Couvidat, F., and Bessagnet, B. (2016). Observations and regional modeling of aerosol optical properties, speciation and size distribution over northern africa and western europe. *Atmospheric Chemistry and Physics*, 16(20):12961–12982.

[Omar et al., 2010] Omar, A., Winker, D. M., Vaughan, M. A., Hu, Y., Trepte, C. R., Ferrare, R. A., Lee, K.-P., Hostetler, C. A., Kittaka, C., Rogers, R. R., Kuehn, R. E., and Liu, Z. (2010). The calipso automated aerosol classification and lidar ratio selection algorithm. *Journal of Atmospheric and Oceanic Technology*, 26(10):1994–2014.

[Sofiev et al., 2012] Sofiev, M., Ermakova, T., and Vankevich, R. (2012). Evaluation of the smoke-injection height from wild-land fires using remote-sensing data. *Atmospheric Chemistry and Physics*, 12(4):1995–2006.

[Winker et al., 2010]  Winker, D., Pelon, J., Coakley Jr., J. A., Ackerman, S. A., Charlson, R. J., Colarco, P. R., Flamant, P., Fu, Q., Hoff, R. M., Kittaka, C., Kubar, T. L., Le Treut, H., McCormick, M. P., Megie, G., Poole, L., Powell, K., Trepte, C., Vaughan, M. A., and Wielicki, B. A. (2010). The calipso mission: A global 3d view of aerosols and clouds. *Bulletin of the American Meteorological Society*, 91(9):1211–1229.

[Zeng et al., 2015]  Zeng, G., Williams, J. E., Fisher, J. A., Emmons, L. K., Jones, N. B., Morgenstern, O., Robinson, J., Smale, D., Paton-Walsh, C., and Griffith, D. W. T. (2015). Multi-model simulation of co and hcho in the southern hemisphere: comparison with observations and impact of biogenic emissions. *Atmospheric Chemistry and Physics*, 15(13):7217–7245.